# DarwinLM: Evolutionary Structured Pruning of Large Language Models

## Abstract

Large Language Models (LLMs) have achieved significant success across various NLP tasks. However, their massive computational costs limit their widespread use, particularly in real-time applications. Structured pruning offers an effective solution by compressing models and directly providing end-to-end speed improvements, regardless of the hardware environment. Meanwhile, different components of the model exhibit varying sensitivities towards pruning, calling for *non-uniform* model compression. However, a pruning method should not only identify a capable substructure, but also account for post-compression training. To this end, we propose *DarwinLM*, a method for *training-aware* structured pruning. *DarwinLM* builds upon an evolutionary search process, generating multiple offspring models in each generation through mutation, and selecting the fittest for survival. To assess the effect of post-training, we incorporate a lightweight, multistep training process within the offspring population, progressively increasing the number of tokens and eliminating poorly performing models in each selection stage. We validate our method through extensive experiments on Llama-2-7B, Llama-3.1-8B and Qwen-2.5-14B-Instruct, achieving state-of-the-art performance for structured pruning. For instance, *DarwinLM* surpasses ShearedLlama while requiring $5\times$ less training data during post-compression training. We also extend our method to MoE models like Qwen3-30B-A3B. To the best of our knowledge, this is the first work to explore non-uniform structured pruning in MoE architectures. Our approach, *DarwinLM*, outperforms uniform pruning baselines and demonstrates the effectiveness of structured sparsity even in complex expert-based models. Code and weights are available.

## 1 Introduction

The high accuracy of Transformer-based models on a wide range of tasks comes with massive computational requirements, which hinders deployability. Thus, there is a line of research focusing on the computational efficiency of Transformer-based models, and in particular large language models (LLMs) via methods such as quantization (Frantar et al., 2022; Dettmers et al., 2023), pruning (Xia et al., 2024; Frantar & Alistarh, 2023) and distillation (Hsieh et al., 2023).

We explore *structured pruning* of LLMs (Molchanov et al., 2017), which works by removing whole rows or columns in the weight matrix, resulting in regular but "thinner" tensors. As such, this approach is orthogonal to "fine-grained" methods such as unstructured pruning and quantization, which can be applied complementarily, and has the advantage that models produced by it can be run faster on mainstream hardware without specific support for low-bit or sparse formats.

In this paper, we provide a new state-of-the-art algorithm for *non-uniform structured pruning with compression guarantees*. Specifically, in *non-uniform* pruning, we leverage the fact that layers or blocks can be compressed to different levels, depending on their sensitivity; in turn, this can be leveraged for higher compression while preserving accuracy (Yin et al., 2023; Sieberling et al., 2024). Second, our algorithm is designed to provide guarantees in terms of the speed or size of the compressed model. While smaller-scale methods such as ZipLM (Kurtić et al., 2024) were able to achieve this for BERT-type models, there are several challenges when extending this to LLMs: for instance, ZipLM only considers the local layer-wise error during the search, which is not consistent with performance on in-context learning (ICL) or downstream tasks, and does not take fine-tuning recovery into account as a metric.

**Contributions.** Our algorithm, called *DarwinLM*, introduces a new evolutionary search approach specifically tailored to structured pruning of LLMs. *DarwinLM* works in two stages: the *search stage*, and the *fine-tuning stage*. The *search* starts from a "parent" model, generated by pruning the original model using second-order information. In each search step, *DarwinLM* generates "offspring" candidate models by copying the parent and "shifting" sparsity from one layer to another, by what we call a *level switch mutation*. Moreover, a central innovation of our approach is that our search process is *fine-tuning aware*: we use a small-scale dataset to briefly fine-tune generated offspring, and select the best offspring after fine-tuning. Once search completes, the *fine-tuning stage* trains the candidate over a small subset of e.g. 10B tokens, after which we perform the final evaluation. Both of these stages are very efficient by design: the pruning and search complete in 8 hours on 4 consumer-grade GPUs, while the LLM fine-tuning completes in half a day on a standard-sized cluster.

In terms of experiments, we scale our method to LLMs of up to 70B parameters (Table 11) from the Llama (Touvron et al., 2023) and Qwen (Qwen, 2024) model families, for which we achieve state-of-the-art performance in one-shot structured pruning by large margins, and match or outperform the performance of comparable prior methods during fine-tuning, while using a very small training budget. Specifically, one-shot pruning results clearly show the superiority of *DarwinLM* relative to prior work, specifically ZipLM Kurtić et al. (2024), ShearedLlama Xia et al. (2024), and EvoPress Sieberling et al. (2024), as well as the Minitron (Sreenivas et al., 2024) and Flextron concurrent work (Cai et al., 2024): for example, when pruning Llama-3.1-8B to half its size, our approach has 5.9% higher average zero-shot accuracy relative to the best prior method (ZipLM).

This major gain in one-shot accuracy enables us to recover good accuracy using much shorter fine-tuning runs relative to competing methods. For instance, in our standard setting we use only 10B tokens for fine-tuning, and are able to reach $> 90\%$ zero-shot accuracy recovery while halving the size of Llama-2-7B. Consequently, we obtain higher accuracy than all prior methods at the same training budget. Moreover, we are able to outperform the ShearedLlama model in terms of accuracy at the same size, even though this model is trained on 5x more tokens (50B). Further, we also compare our method with the line of coarser-grained structured pruning methods including ShortGPT (Men et al., 2024), Shortened-Llama (Kim et al., 2024), and EvoPress (Sieberling et al., 2024) in a one-shot setting, showing that *DarwinLM* provides better performance across compression rates.

To further showcase the flexibility and performance of *DarwinLM*, we demonstrate it to be directly applicable to mixture-of-experts (MoE) models. Specifically, provide an extension of *DarwinLM* to perform one-shot pruning of the recent Qwen-3 MoE with 30B total parameters, out of which 3B are activated per token. We create a smaller accurate variant in one-shot with 20B total parameters, out of which 2B are activated, which retains $\geq 90\%$ of the accuracy of the base model. Moreover, with 10B token finetuning, a compressed 16B variant can also achieve $\geq 90\%$ of the accuracy of the original model. As such, *DarwinLM* is the first structured pruning method to show good results for MoE.

## 2 RELATED WORK

**Structured Pruning Methods.** Structured pruning methods for LLMs (Ma et al., 2023; Men et al., 2024; Kim et al., 2024) typically focus on pruning along the depth dimension or on pruning width (such as attention heads, and MLP intermediate dimensions). Among recent advances, the state-of-the-art is provided by ShearedLLaMA (Xia et al., 2024), which utilizes targeted structured pruning, which reduces a larger model to a specified target shape by pruning layers, attention heads, and intermediate or hidden dimensions in an end-to-end process that is split into regularized fine-tuning, pruning, and further fine-tuning. In addition, it implements *dynamic batching*, which adjusts the composition of sampled data in each training batch, based on varying loss proxies across evaluation domains. By comparison with ShearedLLaMA, *DarwinLM* provides more accurate structured pruning, combining evolutionary search and second-order information. Our results show that our method requires only a fraction of the data to recover accuracy. At the same time, our approach is compatible with their dynamic batching, and should benefit from it. For MoE models, He et al. (2024) explored unstructured and block drop in MoE models while Li et al. (2025) prunes the experts uniformly and applies KD to recover the performance. The recent work on Minitron (Muralidharan et al., 2024) and Flextron (Cai et al., 2024) connected NAS with structured pruning, by establishing a set of effective compression practices for pre-trained LLMs by integrating depth and width pruning with knowledge distillation (KD)-based retraining. These practices are derived from an in-depth empirical

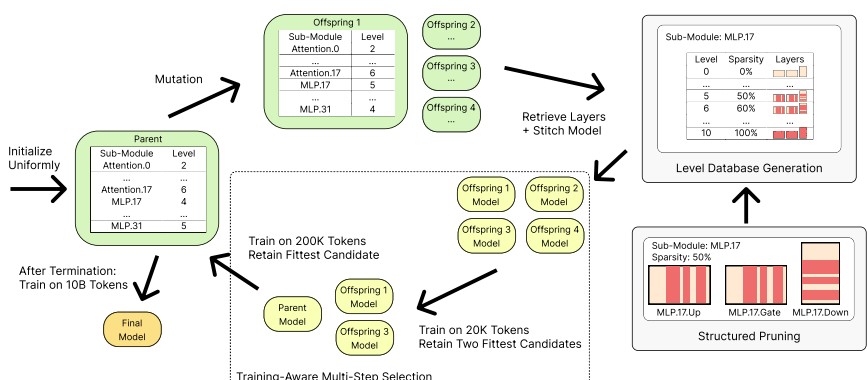

Figure 1: Visual illustration of the *DarwinLM* pipeline.

exploration of pruning strategies across each axis, methods for combining different axes, distillation approaches, and pruning techniques to identify an optimal compressed model. Our contributions are orthogonal to the training strategy proposed in Minitron and Flextron, as we mainly investigate more accurate pruning techniques—many of their findings should also transfer to our setting, and our pruning technique can be applied in their setting. Unfortunately, these approaches use a closed fine-tuning dataset, which prevents us from comparing models end-to-end. In Tables 1 and 2 we provide individual task comparisons; specifically, the latter shows that our one-shot pruning approach is considerably better than Minitron, outperforming it by 15% accuracy on average.

**Non-uniform Pruning Methods.** The distribution of importance across depth, attention heads, and width in the model varies between layers and is not uniform. Low-importance modules tend to be concentrated in specific locations and positions within the model. In the LLM domain, Klein et al. (2023) utilized multi-objective NAS to compress LLMs while optimizing their performance for fine-tuning downstream tasks. SIMPLE (Tao et al., 2023) detects redundant network structures by applying learnable masks to all compressible components, followed by sparse training. EvoPress (Sieberling et al., 2024) performs an evolutionary optimization procedure for *non-uniform unstructured pruning*, *non-uniform quantization*, and *layer dropping*, with a focus on achieving a target model size in a *one-shot setting*. By contrast, *DarwinLM* builds upon fine-grained structured pruning (at the level of rows/columns), optimizes compression allocation under a hardware-specific speedup constraint, and incorporates the effect of continued training into the fitness evaluation of the evolutionary search. The more fine-grained structured pruning we employ significantly improves performance, while guaranteeing speedups without specific hardware support (contrary to e.g. unstructured sparsity). Additionally, two equally performing pruned models can respond differently to continued training, which motivates integrating a lightweight finetune into the search process.

**Other Compression Methods.** Several approaches have been explored in the literature to reduce computational and memory requirements of LLMs without significantly degrading performance, including knowledge distillation, quantization, binarization, and sparsity. In knowledge distillation (Hinton et al., 2015; Sanh, 2019; Gu et al., 2024; Liu et al., 2024; Xu et al., 2024a), a smaller, simpler model (the "student") is trained to replicate the behavior of a larger, more complex model (the "teacher"). The goal is to transfer the knowledge from the teacher to the student while retaining most of the performance benefits of the larger model. Quantization (Xiao et al., 2023; Lin et al., 2024; Li et al., 2024b; Wang et al., 2023; Huang et al., 2024; Xu et al., 2024b; Ma et al., 2024; Tang et al., 2024) reduces the precision of model weights and activations. While this can dramatically reduce the model size and computation, the challenge lies in maintaining accuracy. Another related research area is neural architecture search (NAS) Liu et al. (2021). Instead of focusing on the architecture module search, our method searches the allocated sparsity for each layer and keeps the search efficient without massive re-training, which is generally required by NAS.

## 3 METHOD

Given a compression target such as sparsity ratio or speedup, *DarwinLM* aims to find the model with the best sparsity allocation adhering to this constraint. Formally, let $s(\cdot)$ be a function measuring the

overall sparsity (or inference time) of a given model, and let $\boldsymbol{T}$ denote the targeted sparsity ratio (or speedup). Then, our problem is reduced to

$$\hat{\boldsymbol{M}} = \arg\max_{\boldsymbol{M}} f(\boldsymbol{M}) \quad \text{s. t.} \quad s(\boldsymbol{M}) \leq \boldsymbol{T}, \tag{1}$$

where $M$ is obtained by first structurally pruning the base model and then performing an additional training stage, and $f(\cdot)$ evaluates the quality of a model. Equation (1) presents a non-differentiable optimization problem and, as such, cannot be optimized with standard first-order methods. Instead, we approach this problem by designing a zeroth-order optimization procedure based on evolutionary search. However, this approach comes with fundamental efficiency challenges: evaluating a single compression profile requires pruning the base model, retraining the pruned model to recover performance, and then computing the quality function $f(\cdot)$. This process may have to be repeated several times, depending on the convergence speed of the evolutionary search.

In the following sections, we present how each of these challenges is addressed in the *DarwinLM* pipeline. Section 3.1 details our evolutionary optimization procedure, which allows efficient optimization of Equation (1). For this purpose, we make use of a precomputed sparse layer database, which is described in Section 3.2. An overview of the pipeline is provided in Figure 1.

## 3.1 EVOLUTIONARY SEARCH

Our approach builds upon the evolutionary search framework, which we tailor to the problem formulation. We provide a step-by-step description below, and pseudocode in the Appendix.

**Fitness Environment.** Although models are typically evaluated based on their performance on downstream tasks, this approach is impractical in our context due to the lengthy evaluation times and the risk of overfitting. As an alternative, we adopt the Kullback-Leibler (KL) divergence between the outputs of the dense model and sparse model on a small calibration dataset as a metric to evaluate the fitness of a candidate. KL divergence is well-established, and has been found to be robust with little data compared to measuring perplexity (Sieberling et al., 2024). Consequently, we rewrite our objective function (1) as

$$\hat{\boldsymbol{M}} = \arg\min_{\boldsymbol{M}} D_{KL}(\boldsymbol{M}) \quad \text{s. t.} \quad s(\boldsymbol{M}) \leq \boldsymbol{T}. \tag{2}$$

**Search Space.** First, we perform one-shot compression of the base model using second-order information, as we will outline in Section 3.2. The employed method has the advantage that it operates per subblock (meaning per MLP or attention), allowing for pre-computing a layer database, and stitching together models with arbitrary non-uniform sparsity. To this end, we retain subblocks with varying but identical sparsity levels to better capture the structural diversity. A more detailed description of the pruning algorithm and database generation is presented in Section 3.2. We then search over this database by searching over lists, where each entry describes the discretized sparsity level of the corresponding subblock. Note that based on the different targets, increasing the sparsity level corresponds to a fixed inference time acceleration or a fixed increase in sparsity.

**Initialization.** Throughout the search process, we only maintain a single model as our population. This is based on the expectation that the fittest model so far is most likely to produce even fitter offspring. Initially, our search algorithm starts from 'uniform' compression, which in the case of a speedup objective means that each subblock has sparsity corresponding to the targeted speedup factor. Then, we can generate offspring by slightly increasing and decreasing sparsity levels of the parent model, as we will describe in the next paragraph. In the case of gradual pruning, we compute the residual value between the target sparsity level in different stages and randomly add the residual value to the results from the previous stage.

**Mutation Process.** In each generation, offspring are generated by first copying the parent configuration, and then applying our mutation operator. First, we sample the number of mutations, which we constrain to be very small. For every mutation, we then sample whether to mutate MLPs or attention modules, which means the mutation only happens in the same blocks. The mutation is then performed by randomly selecting one unit to decrease sparsity, and another to increase sparsity. Therefore, we never swap sparsity levels between an attention and an MLP module. Since we designed the database generation in such a way that the difference between two sparsity levels always corresponds to a fixed

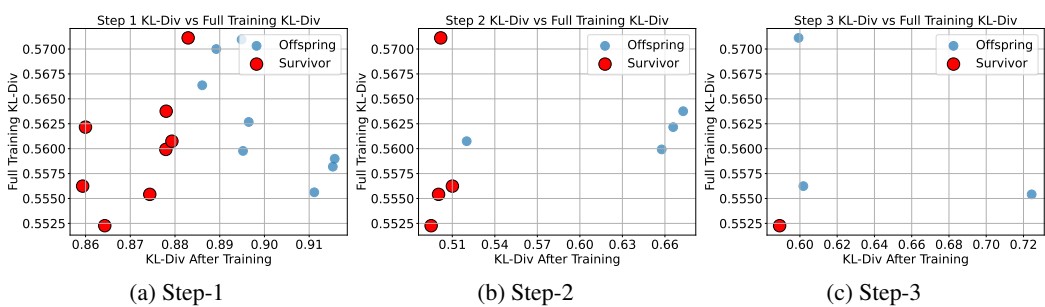

Figure 2: Motivation for training-aware selection. The Y-axis depicts the KL-Divergence of the model after training on $2M$ tokens, while the x-axis is the KL-Divergence after training on a much smaller dataset ($10K, 50K, 200K$ tokens respectively).

sparsity difference, increasing the sparsity level at one subblock and decreasing the sparsity level at another subblock maintains the targeted sparsity ratio.

**Multi-step Training-aware Selection Process.** Our goal is not only to find the best sparse model in a one-shot setting, but to account for continued training. We start from the observation that training on a small amount of data is a good predictor of larger-scale fine-tuning performance. We demonstrate this in Figure 2, where we generate 16 offspring for Llama2-7B. We first use 2M tokens to train all offspring as a "large-scale" full training. Ideally, we want to exclude poorly performing offspring early in the selection process, before spending significant resources on continued training. Therefore, we apply 3 selection steps, each with $[8, 4, 1]$ survivors respectively. In the first step, all offspring are trained on only $10K$ tokens, which is drastically increased to $50K$ and $200K$ in the second and third selection steps. As depicted in Figure 2, the best offspring after full finetuning is successfully identified in the selection process. This motivates our approach, which we term *training-aware offspring selection*, a method that incorporates lightweight finetuning into the selection process, applied in a multi-step manner. Specifically, the training and selection are performed iteratively over $S$ rounds. In each round, a progressively smaller subset of offspring is retained, while the number of samples for training as well as fitness evaluation is increased. The final surviving candidate is selected as the starting point for the next generation.

### 3.2 PRUNED LAYER DATABASE

In this section, we first discuss pruning a specific layer to a given sparsity using second-order information. Then, we introduce how the sparsity level database is generated, which forms the basis of the evolutionary search.

**Second-Order Structured Pruning.** Pruning based on second-order information was first introduced in Optimal Brain Surgeon (OBS) (Hassibi & Stork, 1992), and has since been adapted to Large Language Models by reducing the problem to a layerwise formulation (Kurtic et al., 2022; Frantar & Alistarh, 2023). We adopt this formulation for *layer-wise structured pruning*, in line with prior work (Kurtić et al., 2024). Specifically, for each layer, given a calibration dataset $\mathbf{X}$ of layer inputs and the original layer weights $\mathbf{W}$, we aim to find

$$\arg\min_{\hat{\mathbf{W}}} ||\mathbf{W}\mathbf{X} - \hat{\mathbf{W}}_{:,\mathbf{M}}\mathbf{X}||_2 \tag{3}$$

subject to $\hat{\mathbf{W}}_{:,\mathbf{M}} \in \mathcal{C}$, where $\mathbf{M}$ refers to a *column mask* and $\mathcal{C}$ is the compression constraint. To ensure that the sparse weights $\hat{\mathbf{W}}$ produce outputs similar to those of the original weights $\mathbf{W}$, we must not only identify the less significant structures for pruning, but also compute an update $\delta$ for the remaining weights to compensate for the error introduced by pruning. For this purpose, denote by $\mathbf{H} = \mathbf{X}\mathbf{X}^T$ the Hessian matrix for the $\ell_2$-minimization problem in Equation 3. Define $\mathbf{W}_{i,\mathbf{M}}$ as the weights in row $i$ masked by $\mathbf{M}$ and let $(\mathbf{H}^{-1})_{\mathbf{M},\mathbf{M}}$ be the submatrix of the inverse Hessian corresponding to the entries under the mask $\mathbf{M}$. Now, we can compute the optimal structured mask with corresponding weight updates $\delta$ by:

$$\arg\min_{\mathbf{M}} \sum_{i=1}^{d_{row}} \mathbf{W}_{i,\mathbf{M}} \cdot (\mathbf{H}_{\mathbf{M},\mathbf{M}}^{-1})^{-1} \cdot \mathbf{W}_{i,\mathbf{M}}^{\mathrm{T}}; \quad \delta = -\mathbf{W}_{:,\mathbf{M}} \cdot (\mathbf{H}_{\mathbf{M},\mathbf{M}}^{-1})^{-1} \cdot \mathbf{H}^{-1}{}_{\mathbf{M},:} \tag{4}$$

This formulation extends the derivation of OBS to account for all rows $d_{row}$. In our context, we focus on two types of pruned structures: (1) head pruning in multi-head self-attention, and (2) pruning of the intermediate dimension of MLP modules.

**Granularity.** To reduce the required memory for storing the database, we enforce the number of pruned dimensions in the MLP modules to be a multiple of $m = 32$. For attention modules, we prune on a per-head basis. For each module, we only consider identifying the pruned columns of the final output matrix, referring to the down projection in the case of an MLP. Once the pruned structure of the output matrix is determined, the corresponding rows are pruned in the other matrices (i.e., the K, Q, and V matrices in the attention module, and the up and gate projections in the MLP). However, if the model applies group-query attention (GQA) (Ainslie et al., 2023), such as in Llama-3.1 and Qwen-2.5, we avoid pruning the K and V matrices. During the forward pass, we remove the corresponding heads in the repeated K and V matrices to obtain computationally compatible structures and reduce computation.

**Level Database Generation.** After generating the initial layer database as described above, we process it to obtain the final sparsity level database used for the evolutionary search. This processing step is required to ensure that all considered models in the search process adhere to the targeted inference acceleration. This is achieved by initializing the search with a valid model and then applying a sparsity-preserving (or speedup-preserving) mutation operator. To this end, the sparsity level database is constructed so that the (absolute) difference in inference time between adjacent levels is consistent across all levels and modules. Inference times are measured on a specific hardware setup using a small calibration dataset. (In our implementation, all attention / MLPs employ the same step size, but the step size for attention differs from that of MLPs.) Thus, we can mutate a model while maintaining the targeted sparsity or inference acceleration by simply increasing the same number of levels as we decrease.

### 3.3 Extension to MoE Architectures

Besides dense models, we further extend *DarwinLM* to Mixture of Experts (MoE) models. Typically, each layer of an MoE model includes an attention module and an MoE block, which consists of a number of MLPs (called experts). Since MoE models are already optimized for efficient inference, we instead focus on reducing the memory requirements by optimizing under a sparsity constraint. In our MoE experiments we omit pruning the attention module since the majority of parameters are located in the expert MLPs. First, each expert is pruned to various sparsity levels and stored in the database. In the rare event that some experts are not activated by any calibration tokens, we apply standard magnitude-based weight pruning as a fallback strategy. After that, we employ the evolutionary search within each expert MLP, and therefore keep uniform sparsity across MoE blocks.

## 4 Experiments

### 4.1 Setup

**Models and Datasets.** Given a target sparsity level and a set of pre-trained weights, our method searches for combinations of per-layer sparsity levels under the sparsity constraint, based on a small generic calibration set. In our experiments, for dense models, we consider Llama-2-7B (Touvron et al., 2023), Llama-3.1-8B (Dubey et al., 2024) and Qwen-2.5-14B-Instruct. For MoE pruning, we apply *DarwinLM* on the Qwen3-30B-A3B model. We also test our method on Moonlight-16B-A3B, which can be found in the Appendix. We utilize the open-source dataset Fineweb-Edu (Lozhkov et al., 2024) for both calibration and post-training. The dataset is filtered according to the sample score provided with the dataset. All samples with a lower score than 0.9 are removed from the dataset, resulting in a dataset with 80B tokens. For the search process, we use at most 16 sequences for calibration, making this process highly lightweight. The finetuning data for the offspring models is also sampled from the Fineweb-Edu dataset. For Qwen3-30B-A3B model, we also use our proprietary high-quality dataset to finetune the compressed model.

**Baselines.** First, we compare our non-uniform sparse model with a uniform sparse model under a similar computational budget. Additionally, on Llama-2-7B, we conduct comparisons with ZipLM (Kurtić et al., 2024), ShearedLlama (Xia et al., 2024) and Minitron Muralidharan et al. (2024). Moreover, we also compare with LoRAP Li et al. (2024a), DISP-LLM Gao et al. (2024) and Flextron

Table 1: Comparison of main results for *DarwinLM* and baseline methods on LLaMA-2-7B. Our method achieves the best average performance on benchmarks compared to the baseline methods. With only 10B tokens of fine-tuning, our method beats ShearedLlama, which is fine-tuned with 50B tokens. (†) refers to training on the same data we use.

| Method (fine-tuning budget) | Param. | SciQ | PIQA | WG | ArcE | ArcC | HS | LogiQA | BoolQ | Avg |
|---|---|---|---|---|---|---|---|---|---|---|
| Dense | 6.7B | 93.7 | 78.1 | 69.3 | 76.4 | 53.0 | 78.6 | 30.7 | 77.7 | 69.2 |
| Uniform (one-shot) | 3.4B | 44.1 | 57.1 | 53.3 | 33.5 | 32.2 | 27.3 | 25.0 | 49.0 | 40.1 |
| LoRAP (one-shot) | 2.7B | 51.2 | 57.2 | 47.9 | 31.3 | 26.3 | 30.0 | 27.5 | 61.9 | 41.6 |
| DISP-LLM (one-shot) | 3.3B | - | 68.3 | 56.2 | 51.1 | 30.2 | 49.3 | - | - | - |
| ZipLM (one-shot) | 4.0B | 87.4 | 64.4 | **58.3** | 53.2 | 33.6 | 50.1 | 25.5 | 63.6 | 54.5 |
| ShearedLLaMA (one-shot) | 2.7B | 84.5 | 66.4 | 53.4 | 49.8 | 28.4 | 47.6 | 27.6 | 50.9 | 51.0 |
| *DarwinLM* (one-shot) | 2.7B | **85.6** | **70.8** | 55.8 | **63.3** | **38.1** | **53.2** | **28.5** | **62.7** | **57.2** |
| Flextron (90B) | 3.4B | - | 74.1 | 62.0 | 66.5 | - | 68.5 | - | - | - |
| ShearedLLaMA (50B) | 2.7B | 90.8 | 75.8 | 64.2 | 67.0 | 41.2 | 70.8 | 28.2 | 63.0 | 62.6 |
| ShearedLLaMA (10B†) | 2.7B | 92.0 | 73.6 | 63.1 | 69.8 | 42.0 | 64.4 | 29.0 | 62.1 | 61.9 |
| ShearedLLaMA (30B†) | 2.7B | 90.3 | 74.7 | 64.0 | 71.4 | 45.1 | 66.9 | 27.2 | 64.5 | 63.0 |
| *DarwinLM* (10B) | 2.6B | **90.8** | 72.2 | **65.1** | 68.5 | **45.0** | 67.2 | 28.5 | **64.6** | **62.8** |

Cai et al. (2024) for reference. ZipLM employs dynamic programming to search for the sparse model structure, while ShearedLlama learns pruning masks for Llama-2-7B's weights and applies large-scale fine-tuning on 50B tokens. We perform the evaluation using the publicly available weights after pruning and fine-tuning, as provided by the respective papers. For ZipLM, we reproduce their implementation at a larger scale, following the original paper's methodology. We limit our comparison with ShearedLlama to Llama-2-7B, as the original paper only reports results for this model, and the tuning costs for adapting it to other models are substantial. We also compare *DarwinLM* in a one-shot setting against other one-shot structured pruning methods, including EvoPress (Sieberling et al., 2024), ShortGPT (Men et al., 2024), and Shortened Llama (Kim et al., 2024). For MoE models, since our work emphasizes the pruning strategies applied to MoE models and their impact on model structure and sparsity, rather than their full post-pruning performance, we only provide the one-shot pruning results. All of these methods perform structured pruning on a per-module or per-layer level. We use the official pre-trained weights released on Huggingface for evaluation.

**Evaluation.** We follow ShearedLlama (Xia et al., 2024) to evaluate our method on several downstream tasks including 0-shot accuracy on ARC-easy (Clark et al., 2018), LogiQA (Liu et al., 2020), PIQA (Bisk et al., 2020), SciQ (Welbl et al., 2017), BoolQ (Clark et al., 2019), 5-shot on MMLU (Hendrycks et al., 2020) and WinoGrande (Sakaguchi et al., 2021), 10-shot on HellaSwag (Zellers et al., 2019) and 25-shot on ARC Challenge (Clark et al., 2018). We utilize the *lm-evaluation-harness* framework (Gao et al.) to evaluate all downstream tasks.

**Implementation Details.** When generating the sparsity level database, we set the minimum and maximum levels to 0 and 10, which indicate 0% and 100% sparsity respectively. On Llama-2-7B, we first prune the model with a target sparsity level 5 in the one-shot setting using 2048 calibration samples and fine-tune the sparse model on 10B tokens. After that, we continue to prune the model to target sparsity level 6 based on the fine-tuned model with 2K calibration data. We prune Llama-3.1-8B and Qwen-2.5-14B-Instruct models with target sparsity level 5. The final pruned models are trained on an additional 10B Fineweb tokens. For the evolutionary search, we set the number of generations to 200. For each generation, we generate $\lambda = 16$ offspring for selection. During selection, we apply 4-step selection with $[1024, 2048, 4096, 8192]$ tokens for fitness computation and $[10K, 50K, 100K, 200K]$ tokens for offspring finetuning. The learning rate for training during the search is 1e-5. The pruning and search process is conducted on a $10\times$ L40 GPU workstation. Our training code is based on the LLM-Foundry repository. Our batch size is 1,024 for Llama-2, 1152 for Llama-3.1, and 2048 for Qwen-2.5. The base learning rate is 1e-4 with a cosine decay strategy.

## 4.2 Main Results

**Results on Dense Models.** We prune three representative dense models including Llama-2-7B, Llama-3.1-8B and Qwen-2.5-14B-Instruct. We prune the Llama-2-7B model down to 2.7B with a target level 6. The main results are shown in Table 1. For the pruned models, our method achieves the highest performance on all downstream tasks, except for WinoGrande, where the ZipLM includes many more parameters. Our method also attains the highest average score. In contrast, the uniform pruning method results in a significant performance drop, with an average accuracy of only 40.1,

Table 2: Comparison of results for *DarwinLM* and baseline models on Llama-3.1-8B. With similar speedup, our method achieves the best performance on all benchmarks compared to baseline methods. After post-training with 10B tokens, the performance recovers from 51.6 to 63.7.

| Model | Method | Param. | SciQ | PIQA | WG | ArcE | ArcC | HS | LogiQA | BoolQ | MMLU | Avg |
|-------|--------|--------|------|------|-----|------|------|-----|--------|-------|------|-----|
| | Dense | 8B | 96.3 | 81.2 | 74.3 | 81.4 | 58.2 | 81.7 | 31.1 | 84.0 | 65.2 | 72.8 |
| Llama-3.1-8B | Uniform | 4.5B | 29.1 | 53.6 | 51.7 | 26.0 | 23.6 | 27.1 | 25.5 | 62.1 | 25.7 | 36.1 |
| | ZipLM | 6B | 65.5 | 60.6 | 56.0 | 40.2 | **36.2** | 34.4 | **28.1** | **63.0** | 27.9 | 45.7 |
| | Minitron | 4.4B | 54.4 | 54.4 | 48.9 | 31.8 | 22.1 | 28.4 | 27.1 | 37.8 | 25.6 | 36.7 |
| | *DarwinLM* (one-shot) | 4.6B | **84.9** | **69.4** | **57.3** | **59.6** | 34.2 | **44.6** | 24.1 | 62.2 | **28.5** | **51.6** |
| | *DarwinLM* (10.0B) | 4.6B | 93.2 | 74.8 | 67.4 | 73.2 | 51.6 | 71.3 | 30.7 | 71.1 | 40.6 | 63.7 |
| | Dense | 14B | 96.8 | 81.9 | 79.1 | 85.7 | 72.8 | 85.1 | 38.5 | 87.9 | 80.0 | 78.6 |
| Qwen-2.5-14B-Instr. | Uniform | 8.6B | 78.2 | 72.7 | 57.6 | **76.1** | 45.6 | 47.0 | 28.1 | 61.6 | 45.5 | 56.9 |
| | ZipLM | 8.5B | 69.0 | 66.4 | 52.8 | 60.1 | 38.3 | 43.3 | 29.6 | 60.2 | 25.0 | 49.4 |
| | Minitron | 8.4B | 88.4 | 59.8 | 51.4 | 45.5 | 23.3 | 33.0 | **32.4** | 67.5 | 36.1 | 48.6 |
| | *DarwinLM* (one-shot) | 8.4B | **84.3** | **73.9** | **60.5** | 75.7 | **48.0** | **53.3** | 29.3 | **66.9** | **43.1** | **59.4** |
| | *DarwinLM* (10.0B) | 8.4B | 89.5 | 78.1 | 70.7 | 79.6 | 57.6 | 74.9 | 33.5 | 73.9 | 57.9 | 68.4 |

Table 3: Comparison of results for *DarwinLM* and baseline models on MoE models.

| Model | Method | Param. | SciQ | PIQA | WG | ArcE | ArcC | HS | LogiQA | BoolQ | MMLU | Avg |
|-------|--------|--------|------|------|-----|------|------|-----|--------|-------|------|-----|
| | Dense | 30B-A3B | 97.0 | 79.7 | 71.5 | 79.7 | 68.6 | 77.8 | 34.7 | 88.8 | 79.6 | 75.2 |
| Qwen-3-30B-A3B | Uniform | 20B-A2B | 95.9 | 75.6 | 65.3 | 75.3 | 59.1 | **60.6** | 31.1 | **84.2** | 64.7 | 67.9 |
| | *DarwinLM* (one-shot) | 19B-A2B | 95.9 | **77.1** | **67.5** | **75.6** | **61.2** | 59.5 | **34.0** | 83.4 | **65.0** | **68.8** |
| | Uniform | 16B-A2B | **94.9** | 71.4 | 60.2 | 73.2 | 52.6 | 47.0 | 33.2 | 75.0 | **55.6** | 62.5 |
| | *DarwinLM* (one-shot) | 16B-A2B | 94.7 | **73.0** | **61.1** | **73.6** | **53.9** | **47.6** | **33.6** | **77.5** | 55.1 | **63.3** |
| | *DarwinLM* (10.0B) | 16B-A2B | 95.9 | 76.2 | 69.4 | 80.4 | 59.0 | 69.9 | 32.5 | 77.0 | 66.9 | 69.7 |

essentially a performance collapse compared to the dense model. Specifically, the uniformly-pruned model generates nearly random results on benchmarks such as WG, HS, LogiQA, BoolQ, and MMLU. By contrast, *DarwinLM* achieves an average score of 57.2, outperforming ZipLM (54.5 with 4.0B parameters) and ShearedLlama (51.0 with 2.7B parameters). This comparison highlights the effectiveness of non-uniform structured pruning, particularly at high sparsity. After post-compression training, the pruned models see a significant recovery in performance. Notably, with only 10B tokens for training, *DarwinLM* reaches an average score of 62.8, surpassing the 62.6 reported by ShearedLlama, which was trained with 50B tokens. Furthermore, when we train the pruned model released by ShearedLlama under the same conditions and with 10B tokens, it achieves an average score of 61.9, which is considerably lower than *DarwinLM*.

We also pruned the Llama-3.1 8B model to 4.6B parameters and Qwen-2.5-14B-Instruct to 8.4B with a target sparsity level 5. The comparison results are shown in Table 2. Similar to Llama-2-7B, the uniformly pruned Llama-3.1-8B model suffers catastrophic degradation. For example, the uniformly pruned model achieves 26.0, 23.6, and 27.1 on ARC-E, ARC-C, and HellaSwag, respectively, close to randomly generated results (25.0%). In contrast, *DarwinLM* significantly improves performance, achieving 59.6, 34.2, and 44.6 on these datasets. Overall, *DarwinLM* shows the best average performance compared to both the uniformly pruned and ZipLM models. After post-compression fine-tuning, *DarwinLM* recovers performance across all benchmarks, with an average score of 63.7. This comparison indicates that, starting from an accurate model, *DarwinLM* can produce competitive models tailored to any runtime/size requirements, at very low training cost.

For Qwen-2.5-14B-Instruct, different from Llama-2-7B and Llama-3.1-8B, the uniformly pruned model of Qwen-2.5 obtains satisfactory performance on all benchmarks with 56.9 on average, surpassing ZipLM with similar sparsity. This indicates the failure case of ZipLM as it only optimizes the local error of pruning. However, *DarwinLM* achieves better than uniform structure. Specifically, *DarwinLM* obtains 59.4 on average on all benchmarks, outperforming the uniform model. After post-compression training with 10B tokens, the performance of *DarwinLM* increases to 68.1.

**Results on MoE Model.** We further extend *DarwinLM* to MoE architectures. We test our method on Qwen-3-30B-A3B model and the results are shown in Table 3. The results show that *DarwinLM* consistently outperforms uniform pruning under equivalent parameter settings. For example, at 19B parameters, *DarwinLM* achieves a 68.8 average, outperforming uniform pruning (67.9), and this advantage holds at 16B as well (63.3 vs. 62.5). After 10B token finetuning, the performance recovers from 63.3 to 69.7. Despite aggressive pruning from the 30B dense model (75.2), our method maintains strong performance, demonstrating the benefit of *DarwinLM* at high sparsity ratios.

Table 4: Speedup and memory analysis of *Darwin­LM* on L40s.

| Model | Throughput (Tokens/s) | Memory (MB) |
|---|---|---|
| Dense 7B | 132.8 | 15296 |
| *DarwinLM* 2.7B | 262.7 (**1.98**× ↑) | 6306 (**2.43**× ↓) |
| Dense 8B | 111.7 | 16870 |
| *DarwinLM* 4.6B | 150.5 (**1.35**× ↑) | 12405 (**1.35**× ↓) |
| Dense 14B | 63.2 | 30297 |
| *DarwinLM* 8.4B | 89.1 (**1.40**× ↑) | 21242 (**1.43**× ↓) |

Table 5: Ablation of our proposed training-aware offspring selection (TAS) on Llama-2-7B with target level 5.

| Model | PIQA | SciQ | ArcE |
|---|---|---|---|
| Uniform | 57.1 | 44.1 | 32.2 |
| *DarwinLM* w/o TAS | 68.8 | 88.2 | 63.5 |
| *DarwinLM* | 69.2 | 88.7 | 63.8 |
| *DarwinLM* w/o TAS + 1B tokens | 73.1 | 91.6 | 69.0 |
| *DarwinLM* + 1B tokens | **74.2** | **92.0** | **70.8** |

### 4.3 ANALYSIS

**Speedup Analysis.** Structured pruning can bring direct runtime speedup and memory reduction without hardware specification. We provide the results of the throughput and memory usage of *DarwinLM* and the corresponding dense model, as shown in Table 4. We evaluated *DarwinLM*'s generation throughput over 20 runs on a single L40s and measured peak memory usage with a sequence length of 4096, batch size 1. Results show that *DarwinLM* consistently outperforms the dense baseline, with improvements roughly proportional to parameter reduction. For instance, *DarwinLM* 2.7B uses 2.43 × less memory and achieves 1.98 × higher throughput—slightly below the ideal due to fixed inference overheads.

**Comparison with One-shot Methods under Different Sparsities.** We further compare *DarwinLM* with several current one-shot structured pruning (layer dropping) methods including EvoPress (Sieberling et al., 2024), ShortGPT (Men et al., 2024), and Shortened Llama (Kim et al., 2024) on Llama-2-7B.

We select 40 samples with 4096 tokens from Fineweb-Edu as the test set and compute the perplexity of each model under different sparsity levels. The comparison results are shown in Figure 3. First, we can observe that even though all pruning methods can preserve performance well under the sparsity of 25%, *DarwinLM* still achieves lower perplexity compared to other one-shot pruning methods. Moreover, the performance of ShortGPT shows dramatic degradation after 25% sparsity while the perplexity of Shortened Llama and *DarwinLM* increases only slightly up to 40% sparsity. However, EvoPress also degrades, reaching a perplexity of more than 30, while *DarwinLM* shows a much more minor degradation for 50% sparsity. Generally, *DarwinLM* outperforms all one-shot methods under different sparsity and maintains stable performance as sparsity increases, demonstrating the effectiveness of our method. This is also natural since our method benefits from higher compression granularity.

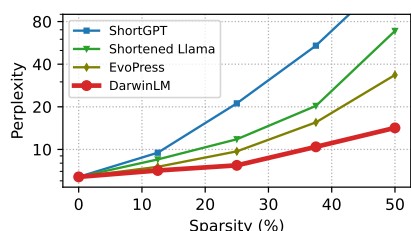

Figure 3: Comparison of *DarwinLM* and other one-shot methods that remove modules entirely. Our method consistently outperforms across all sparsity levels, demonstrating the effectiveness of our finer-grained structured pruning approach. The y-axis is log-scaled.

**Ablation Study.** Finally, we examine the impact of training-aware selection for structure searching and post-training. The results are presented in Table 5. First of all, both models with and without training-aware selection (TAS in the context) searched with 200 generations are better than uniform models. Furthermore, the performance gap of *DarwinLM* with and without TAS is minor before training, indicating that applying TAS generates sparse models with similar performance. However, after 1B tokens of training for each model, the performance gap between models with and without TAS becomes larger, demonstrating that with training-aware selection, *DarwinLM* is able to select a more suitable model for post-training. Full results can be found in the Appendix Table 25.

## 5 CONCLUSION

We introduced a novel non-uniform, training-aware structured pruning method called *DarwinLM*, which generates compressed models by evolutionary search. *DarwinLM* efficiently searches compressed models over a layer database, and incorporates the offspring models' aptitude for continued pretraining into the search procedure. Experiments on Llama-2-7B, Llama-3.1-8B, Qwen-2.5-14B-Instruct and Qwen-3-30B-A3B demonstrate that our approach achieves state-of-the-art performance. *DarwinLM* is remarkably sample-efficient, as it can match or even improve upon the performance of prior methods which required 10x more data and training computation.

## ETHICS STATEMENT

This work improves the efficiency of large language models through structured pruning. Our experiments use only publicly available pre-trained models and open datasets (e.g., Fineweb-Edu) without human subjects or sensitive data. While reducing computation lowers costs and environmental impact, it may also facilitate wider use of LLMs, including potential misuse or amplification of biases. We release our method for research purposes only and encourage responsible use, including alignment and safety checks, before deployment.

## REPRODUCIBILITY STATEMENT

We provide detailed descriptions of our pruning algorithm, evolutionary search procedure, and training setup in the main text and Appendix. All hyperparameters, datasets, and evaluation protocols are specified (except the finetuning data for Qwen3-MoE-30A3B model), and we use widely available pre-trained models (Llama, Qwen). Our code and pruned model weights will be released to ensure full reproducibility of our results.

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

# A APPENDIX

## A.1 USE OF LLMS

In this paper, we use an LLM to help revise and polish the writing of the paper, while all ideas and experiments are conceived and carried out entirely by the authors.

## A.2 THE SEARCH ALGORITHM

---

**Algorithm 1** DarwinLM: Evolutionary optimization with training-aware offspring selection.

---

**Input:**
$N$: number of generations.
$S$: number of selection steps.
$\lambda$: number of offspring in each generation.
$T_f$: list of tokens for finetuning.
$T_s$: list of tokens for selection.

**Initialization:**
$D \leftarrow databaseGen()$
## Sampled levels are all integers
$parent \leftarrow UniformLevelSample()$

**Optimization:**
**for** $t \leftarrow 1$ to $N$ **do**
  ## Elitism
  $candidates \leftarrow [parent]$
  ## Offspring generation via mutation
  **for** $i \leftarrow 1$ to $\lambda$ **do**
    $offspring \leftarrow LevelSwitchMutation(parent)$
    $candidates.append(offspring)$
  **end for**
  ## Multi-step training-aware selection
  **for** $step \leftarrow 1$ to $S$ **do**
    $cand\_models = []$
    **for** $candidate \in candidates$ **do**
      $cand\_model \leftarrow stitch(candidate, D)$
      $cand\_model \leftarrow train(cand\_model, T_f[step])$
      $cand\_models.append(cand\_model)$
    **end for**
    $candidates \leftarrow selectTopKFit(cand\_models, T_s[step])$
  **end for**
  $parent \leftarrow candidates[0]$
**end for**
**return** $parent$

---

## A.3 IMPLEMENTATION DETAILS

**Details of second-order structured pruning.** We utilize 2,048 sequences with 4,096 tokens from the Fineweb-Edu dataset Lozhkov et al. (2024) as calibration data for Llama-2-7B, Llama-3.1-8B, and Qwen-2.5-14B-Instruct. In the attention module, we prune entire attention heads, and in the MLP module we prune entire columns of the output matrix. For Llama-2-7B, we prune the input matrix, as well as the Q, K, and V matrices, based on the pruned output matrix in the attention module. For Llama-3.1-8B and Qwen-2.5-14B-Instruct, which both use grouped query attention, we omit the key and value matrices for pruning. For MoE models, we do not prune the attention module and follow the same experimental setting as Qwen-2.5-14B-Instruct otherwise. For all models, the input and gate matrices in the MLP module are pruned according to the output matrix. Pruning

Llama-2-7B, Llama-3.1-8B, and Qwen-2.5-14B-Instruct requires $4\times$ 48GB GPU memory. Most of the second-order structured pruning experiments are conducted on a $4\times$ NVIDIA L40S machine with 48GB GPU memory.

**Details of the evolutionary search.**    Given a target sparsity level, the search process starts from uniform initialization. During selection, we apply 4 steps of selection with $[1024, 2048, 4096, 8192]$ tokens for fitness computation and $[10K, 50K, 100K, 200K]$ tokens for offspring finetuning. The number of survivors is set to $[8, 4, 2, 1]$ for each step and model. We set the learning rate for offspring training to 1e-5. For Llama-2-7B, we apply gradual pruning with target sparsity level 5 in the first stage. We perform the search procedure for 200 generations. After training on 10B tokens, we search again with target sparsity level 6 (60% sparsity) for 500 generations. For Llama-3.1-8B and Qwen-2.5-14B-Instruct, we search the sparse model with target sparsity level 5 (50% sparsity) for 200 generations. The search process for the 7/8B models can be done on a single GPU with 48GB of memory. Qwen-2.5-14B-Instruct and Moonlight-16B-A3B models are searched with $6 \times$ L40S GPUs. Qwen-3-30B-A3B experiments are conducted on $6 \times$ H100 GPUs.

Table 6: Hyperparameter details for post-training on *DarwinLM*-2.6B, *DarwinLM*-4.4B, and *DarwinLM*-8.4B.

| Parameter | *DarwinLM*-2.6B | *DarwinLM*-4.4B | *DarwinLM*-8.4B | *DarwinLM*-16A2B |
|---|---|---|---|---|
| Learning rate | 1e-4 | 1e-4 | 1e-4 | 2.4e-4 |
| Global batch size | 1024 | 1152 | 2048 | 512 |
| Warm-up steps | 50 steps | 10 steps | 50 steps | 50 steps |
| LR decay scheduler | Cosine | Cosine | Cosine | Cosine |
| Context length | 4,096 | 8,192 | 4,096 | 4096 |
| Overall tokens | 10B | 10B | 10B | 10B |

**Details of post-training.**    We train the final 2.6B sparse model, pruned from Llama-2-7B, and the 4.4B model, pruned from Llama-3.1-8B, on 10B tokens each. Gradient accumulation is used to achieve a larger global batch size. The models are trained with the Adam optimizer, using a learning rate of 1e-4, and a cosine learning rate decay scheduler. No weight decay is applied. The training process is conducted on a cluster of 40 H100 GPUs for 13 hours. Detailed hyperparameters for post-training can be found in Table 6.

## A.4    ADDITIONAL EXPERIMENTAL RESULTS

Table 7: The throughput and latency of *DarwinLM* with vLLM serving framework.

| Model | Throughput (Tokens/s) | Latency (ms) |
|---|---|---|
| LLaMA-2-7B | 2469.57 | 51.83 |
| Shearedllama-2.7B | 4482.95 | 28.55 |
| *DarwinLM*-2.7B | 5675.29 | 22.55 |

**Speed of real-world deployment.**    We provide the inference throughput and latency on vLLM inference framework and also add the comparison with Dense model and Shearedllama (with sequence length of 1024, request number of 128, single L40s GPUs), as shown in Table 7. The results clearly show that the irregular shapes do not affect latency in a negative way. Moreover, we find that DarwinLM achieves higher throughput and lower latency compared with Shearedllama, at a similar parameter count. The reason DarwinLM is faster is that, with our approach, some Attention / MLP blocks are removed completely, which reduces both the computation, and the communication cost between SRAM and HBM inside the GPU. Furthermore, we believe that such a structure will bring extra efficiency benefits in the case of huge models, which require tensor-parallel or pipeline-parallel for inference, since removing a whole block significantly reduces block-wise communication cost.

**Comparison with random search.**    We provide the performance comparison with different searching techniques, including ZipLM and random search, as shown in Table 8. The results show a major accuracy advantage in favor of DarwinLM, with an improvement of almost 4% on average, across tasks, relative to Random search, and even higher relative to ZipLM. This highlights the advantage of our search strategy.

Table 8: Performance comparison with ZipLM and random search.

| Model | Method | Param. | SciQ | PIQA | WG | ARC-E | ARC-C | HS | LogiQA | BoolQ | MMLU | Avg |
|---|---|---|---|---|---|---|---|---|---|---|---|---|
| Llama-3.1-8B | Random Search | 4.6B | 78.1 | 65.5 | 52.3 | 54.5 | 26.2 | 31.6 | 24.1 | 62.1 | 26.5 | 46.7 |
| | ZipLM | 6B | 65.5 | 60.6 | 56.0 | 40.2 | 36.2 | 34.4 | 28.1 | 63.0 | 27.9 | 45.7 |
| | *DarwinLM (one-shot)* | 4.6B | 84.9 | 69.4 | 57.3 | 59.6 | 34.2 | 44.6 | 24.1 | 62.2 | 28.5 | 51.6 |

Table 9: The results of *DarwinLM* with 50B token training.

| Methods | SciQ | PIQA | WG | ARC-E | ARC-C | HS | LogiQA | BoolQ | MMLU | Avg |
|---|---|---|---|---|---|---|---|---|---|---|
| Dense | 97 | 79.7 | 71.5 | 79.7 | 68.6 | 77.8 | 34.7 | 88.8 | 79.6 | 75.2 |
| DeepSeek-MoE-base 16A2B 2T token | 92.9 | 80.5 | 72.7 | 75.9 | 53.2 | 79.9 | 29.1 | 72.9 | 45 | 66.9 |
| DeepSeek-V2-Lite 16A2B 5.7T token | 93.5 | 79 | 69.2 | 75.5 | 51.9 | 74.6 | 29.1 | 74.3 | 48.4 | 66.1 |
| *DarwinLM* 16A2B MoE 10B token | 95.9 | 76.2 | 69.4 | 80.4 | 59 | 69.9 | 32.5 | 77 | 66.9 | 69.7 |
| *DarwinLM* 16A2B MoE 50B token | 96 | 77.1 | 70.1 | 81.9 | 60.5 | 72.5 | 32.7 | 78 | 69.1 | 70.8 |

**Training with more tokens.** We provide the results of *DarwinLM* trained with 50B tokens, as shown in Table 9. With more tokens, the performance of *DarwinLM* continues to improve consistently. Moreover, *DarwinLM* achieves better performance than DeepSeek-MoE-base and DeepSeek-V2-Lite, which are trained with 2T and 5.7T tokens, respectively.

Table 10: Searched sparsity distribution of *DarwinLM*-2.7B including the attention head number and MLP size.

| Type | Value |
|---|---|
| *DarwinLM* Attn Head Num | 25, 21, 18, 18, 14, 10, 14, 10, 18, 14, 18, 0, 0, 28, 21, 10, 18, 14, 10, 18, 10, 10, 10, 14, 0, 0, 14, 1, 4, 0, 6, 0 |
| Shearedllama Attn Head Num | 20 for all layers |
| *DarwinLM* MLP Size | 3104, 8032, 6496, 4256, 5280, 5280, 4256, 3104, 5280, 4256, 1824, 0, 3104, 5280, 5280, 4256, 4256, 6496, 6496, 5280, 3104, 4256, 4256, 3104, 3104, 3104, 4256, 3104, 3104, 3104, 6496, 6496 |
| Shearedllama MLP Size | 6912 for all layers |

**Searched sparsity distribution.** We provide the searched sparsity distribution of *DarwinLM* in Table 10.

**Results on large-scale models.** Table 11 compares one-shot pruning methods on Llama-3.1-70B. The full dense model (70B params) achieves the highest average score (78.8). Uniform pruning (35B) drops to 73.9, while *DarwinLM* (35B) improves to 75.0, outperforming uniform pruning across most benchmarks. *DarwinLM* preserves performance better, especially on ArcE, HS, and BoolQ, suggesting the effectiveness of *DarwinLM*.

**Results on more models.** Besides scaling up the method to large models, *DarwinLM* is also applied to small-scale models, such as Pythia-2.8B and Gemma2-2B, as shown in Table 12. At this scale, without any finetuning, *DarwinLM* achieves downstream performance that is remarkably close to that of the larger dense model. Table 12 presents a comparison across multiple benchmarks, where *DarwinLM*, using only 1.4B parameters (half the size of the dense model), consistently outperforms the uniform baseline and performs competitively with the dense model. Notably, *DarwinLM* surpasses the dense model on tasks like BoolQ (65.0 vs. 64.5), shows near-parity on ArcE (61.2 vs. 64.4), and delivers strong results on SciQ (82.9) and PIQA (71.3). The average performance of *DarwinLM* (53.3) significantly exceeds that of the uniform baseline (47.4) and comes close to the dense model's 55.6. We further provide the results of *DarwinLM* on Mistral-7B model, as shown in Table 13.

**Result comparison with model trained from scratch.** We provide the comparison of *DarwinLM* with open-source models trained from scratch (OLMO and Baichuan2) on multiple benchmarks, as shown in Table 14. Despite using fewer training tokens than some baselines, *DarwinLM* achieves competitive or superior performance. Notably, *DarwinLM* 8.4B (10B tokens) outperforms both OLOMO 7B (2T) and Baichuan2 7B (2.6T), achieving a higher average score (68.4 vs. 67.9 and 66.4). It excels particularly on ArcE (79.6) and LogiQA (33.5), indicating strong reasoning capabilities. The 4.6B *DarwinLM* also matches or exceeds OLOMO 7B in most metrics despite smaller size.

**Post-training with LoRA.** Besides full finetuning, our model can also be finetuned with parameter-efficient finetuning techniques such as LoRA (Hu et al., 2022). We provide the results in Table 15.

Table 11: Results on Llama-3.1-70B. We omit training and report one-shot pruning performance.

| Model | Methods | Param. | SciQ | PIQA | WG | ArcE | ArcC | HS | LogiQA | BoolQ | Avg |
|---|---|---|---|---|---|---|---|---|---|---|---|
| Llama-3.1-70B | Dense | 70B | 96.5 | 82.9 | 85.2 | 87.2 | 69.3 | 87.8 | 37.0 | 85.2 | 78.8 |
| | Uniform | 35B | 95.1 | 80.1 | 81.7 | 80.8 | 59.9 | 78.2 | 33.1 | 82.3 | 73.9 |
| | *DarwinLM* (one-shot) | 35B | 95.4 | 81.2 | 83.5 | 82.5 | 60.5 | 80.3 | 33.0 | 84.2 | 75.0 |

Table 12: Results on Pythia-2.8B and Gemma2-2B, which include less model parameters and more model families. Here, we omit continued training, and report the one-shot pruning performance.

| Model | Methods | Param. | SciQ | PIQA | WG | ArcE | ArcC | HS | LogiQA | BoolQ | MMLU | Avg |
|---|---|---|---|---|---|---|---|---|---|---|---|---|
| Pythia-2.8B | Dense | 2.8B | 88.3 | 73.8 | 58.6 | 64.4 | 35.8 | 60.1 | 28.5 | 64.5 | 26.7 | 55.6 |
| | Uniform | 1.4B | 75.9 | 59.4 | **59.1** | 39.0 | 29.1 | 50.3 | 25.9 | 62.6 | **26.0** | 47.4 |
| | *DarwinLM* (one-shot) | 1.4B | **82.9** | **71.3** | 57.3 | **61.2** | **34.7** | **54.5** | 27.9 | **65.0** | 25.1 | **53.3** |
| Gemma2-2B | Dense | 2.5B | 94.6 | 76.7 | 65.2 | 74 | 49.2 | 71.5 | 29.8 | 70.0 | 41.2 | 63.5 |
| | Uniform | 1.2B | 78.7 | 58.1 | 50.5 | 41.0 | 21.0 | 26.4 | 25.1 | 52.7 | 25.6 | 42.1 |
| | *DarwinLM* (one-shot) | 1.2B | 80.0 | 61.3 | 52.1 | 48.5 | 23.2 | 30.5 | 26.4 | 55.5 | 25.3 | 44.7 |

The results show that LoRA can achieve reasonable improvement based on the pruned model while full finetuning obtains better performance given identical tokens.

**Additional results in comparison to uniform pruning.** We present a full comparison of the uniformly pruned models and the sparse models obtained via *DarwinLM* in Table 16. For all three models (Llama-2-7B, Llama-3.1-8B, and Qwen-2.5-14B-Instruct), *DarwinLM* consistently outperforms uniform pruning on evaluation tasks, with immense gains for Llama-2-7B (54.2 vs. 38.4 on average) and Llama-3.1-8B (51.6 vs. 36.1 on average).

**Performance on generation tasks.** We compare our method with SharedLlama on GSM-8K, a generation task. The results are shown in Table 17. While the overall performance is low (as expected for small models without finetuning), *DarwinLM* consistently outperforms ShearedLLaMA under the same data budget, nearly matching its 50B-tokens performance with just 10B tokens.

**Results on additional MoE models.** Besides the Qwen3-30B-A3B MoE model, we also apply *DarwinLM* on Moonlight-16B-A3B, another mixture of experts model. The results are shown in Table 18. Overall, *DarwinLM* obtains a more capable sparse model on downstream tasks compared to uniform pruning.

**Running time comparison.** We compare the running time for pruning with ShearedLlama in Table 19. ShearedLlama has higher computational cost for pruning since it requires additional training to find the weight masks. Additionally, the hardware requirements of *DarwinLM* are lower than that of ShearedLlama.

**Additional results of post-training comparison with ShearedLlama.** We provide the post-training comparison of ShearedLlama across all benchmarks, with the performance trends for each dataset available in Figure 4. Both methods prune Llama-2-7B, with *DarwinLM* producing a model with 2.6B parameters and ShearedLlama producing a model with 2.7B parameters. *DarwinLM* outperforms ShearedLlama on benchmark evaluations in most cases, including SciQ, PIQA, ARC-E, ARC-C, HellaSwag, WinoGrande, LogiQA, BoolQ, and MMLU.

Table 13: Results on Mistral-7B. We omit continued training, and report one-shot pruning performance.

| Model | Methods | Param. | SciQ | PIQA | WG | ArcE | ArcC | HS | LogiQA | BoolQ | MMLU | Avg |
|---|---|---|---|---|---|---|---|---|---|---|---|---|
| Mistral-7B | Dense | 7B | 95.9 | 80.8 | 79.4 | 80.5 | 61.3 | 83.3 | 30.2 | 83.3 | 62.5 | 73.0 |
| | Uniform | 3.9B | 57.2 | 66.4 | 50.5 | 62 | 32.7 | 37.5 | 27.6 | 53.7 | 26.0 | 45.9 |
| | *DarwinLM* (one-shot) | 3.9B | 84.2 | 65.7 | 54 | 57.8 | 34.1 | 38.9 | 26.8 | 60.5 | 26.5 | 49.8 |

Table 14: Result comparison of *DarwinLM* and the open-source model trained from scratch.

| Model (Training token) | SciQ | PIQA | WG | ArcE | ArcC | HS | LogiQA | BoolQ | MMLU | Avg |
|---|---|---|---|---|---|---|---|---|---|---|
| OLMO 7B (2.5T) | 92.8 | 79.4 | 70.4 | 73.3 | 44.9 | 77.1 | 27.9 | 72.5 | 28.3 | 62.9 |
| *DarwinLM* 4.6B (10B) | 93.2 | 74.8 | 67.4 | 73.2 | 51.6 | 71.3 | 30.7 | 71.1 | 40.6 | 63.7 |
| Baichuan2 7B (2.6T) | 94.8 | 77.1 | 72.2 | 75.0 | 49.5 | 73.0 | 28.7 | 73.9 | 54.0 | 66.4 |
| OLMO 0424 7B (2T) | 96.1 | 80.1 | 72.1 | 73.8 | 49.2 | 78.0 | 29.3 | 80.8 | 52.1 | 67.9 |
| *DarwinLM* 8.4B (10B) | 89.5 | 78.1 | 70.7 | 79.6 | 57.6 | 74.9 | 33.5 | 73.9 | 57.9 | 68.4 |

Table 15: Comparison of different training techniques.

| Model (Training Tokens) | SciQ | PIQA | WG | ArcE | ArcC | HS | LogiQA | BoolQ | Avg |
|---|---|---|---|---|---|---|---|---|---|
| *DarwinLM*-2.7B-Pruned | 85.6 | 70.8 | 55.8 | 63.3 | 38.1 | 53.2 | 28.5 | 62.7 | 57.3 |
| Full-Finetuning (10B) | 90.8 | 72.2 | 65.1 | 68.5 | 45.0 | 67.2 | 28.5 | 64.6 | 62.7 |
| LoRA (10B) | 88.2 | 73.2 | 69.4 | 57.2 | 40.6 | 61.4 | 29.1 | 61.6 | 60.0 |

## A.5 ABLATIONS

**Ablation of the search metric.** Here, we compare different fitness functions used during the evolutionary search. As shown in Table 20, we compare using perplexity (PPL) and KL-Divergence (KL-Div) to evaluate the fitness of candidate models. Both metrics yield similar performance on downstream tasks, which demonstrates the robustness of *DarwinLM* to the objective type.

**Ablation of the number of offspring.** We provide an ablation study for varying the number of offspring in Table 21. When the offspring number increases, the downstream performance also improves with the cost of additional searching time. However, the performance seems to plateau beyond 24 offspring per generation. Therefore, choosing a relatively small offspring number for each generation achieves satisfactory performance with acceptable searching cost.

**Ablation of the number of sparsity levels.** Another hyperparameter of *DarwinLM* is the number of sparsity levels in the layer database. We provide results with a higher number of sparsity levels in Table 22. When more sparsity levels are available, *DarwinLM* can search more fine-grained and thus achieve better downstream performance. This comes at the cost of having to store a larger database, and a higher number of generations required for convergence in the search process.

**Ablation of the finetuning tokens.** We provide the ablation of different finetuning token choice on Llama3.1-8B, as shown in Table 23. The average scores are nearly identical—51.6 and 51.6—across both token configurations, with minimal variation across individual benchmarks. This demonstrates that *DarwinLM* is robust to the amount of finetuning data used in the search process, maintaining consistent performance even with significantly fewer tokens

**Ablation of pruning methods** *DarwinLM* can build upon all pruning techniques. To show the effectiveness of *DarwinLM*, we provide the results of model pruned by the simplest pruning method, namely magnitude-based pruning on Llama-3.1-8B. The results are shown in Table 24. We can observe that even with the simplest pruning method, *DarwinLM* can bring benefits to the final results.

**The full results of Table 5.** We further provide the full results of Table 5, as shwon in Table 25.

Table 16: Full comparison of *DarwinLM* with uniform pruning on Llama-2-7B, Llama-3.1-8B and Qwen-2.5-14B-Instruct.

| Model | Method | Param. | SciQ | PIQA | WG | ArcE | ArcC | HS | LogiQA | BoolQ | MMLU | Avg |
|---|---|---|---|---|---|---|---|---|---|---|---|---|
| Llama-2-7B | Dense | 6.7B | 93.7 | 78.1 | 69.3 | 76.4 | 53.0 | 78.6 | 30.7 | 82.1 | 46.6 | 67.6 |
| | Uniform | 3.3B | 44.1 | 57.1 | 53.3 | 33.5 | 32.2 | 27.3 | 25.0 | 49.0 | 23.7 | 38.4 |
| | *DarwinLM* | 3.3B | **89.1** | **70.0** | **59.4** | **63.7** | **36.2** | **53.5** | 25.9 | **65.3** | **24.8** | **54.2** |
| Llama-3.1-8B | Dense | 8B | 96.3 | 81.2 | 74.3 | 81.4 | 58.2 | 81.7 | 31.1 | 84.0 | 65.2 | 72.8 |
| | Uniform | 4.5B | 29.1 | 53.6 | 51.7 | 26.0 | 23.6 | 27.1 | 25.5 | 62.1 | 25.7 | 36.1 |
| | *DarwinLM* | 4.6B | **84.9** | **69.4** | **57.3** | **59.6** | **34.2** | **44.6** | 24.1 | **62.2** | **28.5** | **51.6** |
| Qwen-2.5-14B-Ins. | Dense | 14B | 96.8 | 81.9 | 79.1 | 85.7 | 72.8 | 85.1 | 38.5 | 87.9 | 80.0 | 78.6 |
| | Uniform | 8.6B | 78.2 | 72.7 | 57.6 | 76.1 | 45.6 | 47.0 | 28.1 | 61.6 | 45.5 | 56.9 |
| | *DarwinLM* (one-shot) | 8.4B | **84.3** | **73.9** | **60.5** | 75.7 | **48.0** | **53.3** | **29.3** | **66.9** | 43.1 | **59.4** |

Table 17: Comparison of *DarwinLM* and ShearedLlama on GSM-8K evaluation set on Llama-2-7B.

| Method | GSM-8K |
|---|---|
| Dense | 15.6 |
| ShearedLLaMA-pruned | 1.1 |
| *DarwinLM*-pruned | 1.9 |
| ShearedLLaMA 50B | 3.7 |
| ShearedLLaMA 10B | 2.0 |
| *DarwinLM* 10B | 3.4 |

Table 18: Comparison of *DarwinLM* and uniform pruning on Moonlight-16B-A3B, a mixture of experts model. Here, we do not perform continued training.

| Method | Param. | SciQ | PIQA | WG | ArcE | ArcC | HS | LogiQA | BoolQ | MMLU | Avg |
|---|---|---|---|---|---|---|---|---|---|---|---|
| Dense | 16B-A3B | 96.0 | 79.1 | 75.5 | 84.6 | 62.7 | 81.6 | 37.1 | 80.1 | 70.1 | 74.1 |
| Uniform | 8.7B-A2B | 94.0 | 71 | **61.9** | **76.3** | 44.2 | **52.9** | 30.5 | 65.5 | 51.3 | 60.8 |
| *DarwinLM* (one-shot) | 8.7B-A2B | **95.4** | **71.7** | 61.5 | 76.0 | **45.0** | 50.4 | 30.5 | **70.3** | **51.8** | **61.4** |

Table 19: Running time comparison with ShearedLlama and *DarwinLM*. *DarwinLM* has lower computational cost compared to ShearedLlama.

| Model | Hardware Requirement | Running Time |
|---|---|---|
| ShearedLlama | $8 \times$ A100-80G | 7.4h |
| *DarwinLM* | $4 \times$ L40S-48G | 6.9h |

Table 20: Comparison of *DarwinLM* with different metrics during search on Llama-3.1-8B.

| Method | Param. | SciQ | PIQA | WG | ArcE | ArcC | HS | LogiQA | BoolQ | MMLU | Avg |
|---|---|---|---|---|---|---|---|---|---|---|---|
| PPL | 4.6B | 84.7 | 69.4 | 58.4 | 61.2 | 32.5 | 43.8 | 25.6 | 62.4 | 27.8 | 51.7 |
| KL-Div | 4.6B | 84.9 | 69.4 | 57.3 | 59.6 | 34.2 | 44.6 | 24.1 | 62.2 | 28.5 | 51.6 |

Table 21: Comparison of *DarwinLM* with different number of offspring during search on Llama-3.1-8B.

| Number of Offspring | SciQ | PIQA | WG | ArcE | ArcC | HS | LogiQA | BoolQ | MMLU | Avg |
|---|---|---|---|---|---|---|---|---|---|---|
| 8 | 84.4 | 69.0 | 56.9 | 58.6 | 33.2 | 43.3 | 24.1 | 62.2 | 28.3 | 51.1 |
| 16 | 84.9 | 69.4 | 57.3 | 59.6 | 34.2 | 44.6 | 24.1 | 62.2 | 28.5 | 51.6 |
| 24 | 84.9 | 69.5 | 58.8 | 61.4 | 30.9 | 47.7 | 26.8 | 62.5 | 27.1 | 52.1 |
| 32 | 86.7 | 69.9 | 58.8 | 61.7 | 31.2 | 45.3 | 24.8 | 62.2 | 28.5 | 52.1 |

Table 22: Comparison of *DarwinLM* with different number of sparsity levels produced during database generation.

| Sparsity Level | SciQ | PIQA | WG | ArcE | ArcC | HS | LogiQA | BoolQ | MMLU | Avg |
|---|---|---|---|---|---|---|---|---|---|---|
| 10 | 84.9 | 69.4 | 57.3 | 59.6 | 34.2 | 44.6 | 24.1 | 62.2 | 28.5 | 51.6 |
| 16 | 87.3 | 70.1 | 58.2 | 60.2 | 32.7 | 46.8 | 25.8 | 62.1 | 32.1 | 52.7 |

Table 23: Comparison of *DarwinLM* with different finetuning tokens during search on Llama-3.1-8B.

| Finetuning Tokens | SciQ | PIQA | WG | ArcE | ArcC | HS | LogiQA | BoolQ | Avg |
|---|---|---|---|---|---|---|---|---|---|
| $[10K, 50K, 100K]$ | 84.9 | 69.4 | 57.3 | 59.6 | 34.2 | 44.6 | 24.1 | 62.2 | 51.6 |
| $[5K, 10K, 20K]$ | 85.8 | 69.8 | 56.1 | 60.9 | 33.6 | 43.8 | 25.3 | 61.1 | 51.6 |

Table 24: Ablation of pruning methods on Llama-3.1-8B.

| Method | SciQ | PIQA | WG | ArcE | ArcC | HS | LogiQA | BoolQ | Avg |
|---|---|---|---|---|---|---|---|---|---|
| Uniform | 19.2 | 53.2 | 49.7 | 24.9 | 26.1 | 26.0 | 25.6 | 40.0 | 33.0 |
| *DarwinLM* | 22.1 | 53.6 | 50.6 | 25.6 | 26.6 | 26.2 | 25.7 | 38.8 | 33.7 |

Table 25: Full results of Table 5.

| Method | SciQ | PIQA | WG | ArcE | ArcC | HS | LogiQA | BoolQ | Avg |
|---|---|---|---|---|---|---|---|---|---|
| Uniform | 44.1 | 57.1 | 53.3 | 33.5 | 32.2 | 27.3 | 25.0 | 49.0 | 40.2 |
| *DarwinLM* w/o TAS | 88.2 | 69.1 | 58.6 | 63.5 | 31.7 | 41.4 | 20.1 | 63.0 | 54.5 |
| *DarwinLM* | 88.7 | 69.2 | 59.9 | 63.8 | 32.5 | 40.1 | 22.2 | 65.1 | 55.1 |
| *DarwinLM* w/o TAS + 1B | 91.6 | 73.1 | 59.9 | 69.0 | 34.1 | 47.2 | 22.1 | 68.2 | 58.1 |
| *DarwinLM* +1B | 92.0 | 74.2 | 60.0 | 70.8 | 36.1 | 48.1 | 22.8 | 66.0 | 58.8 |

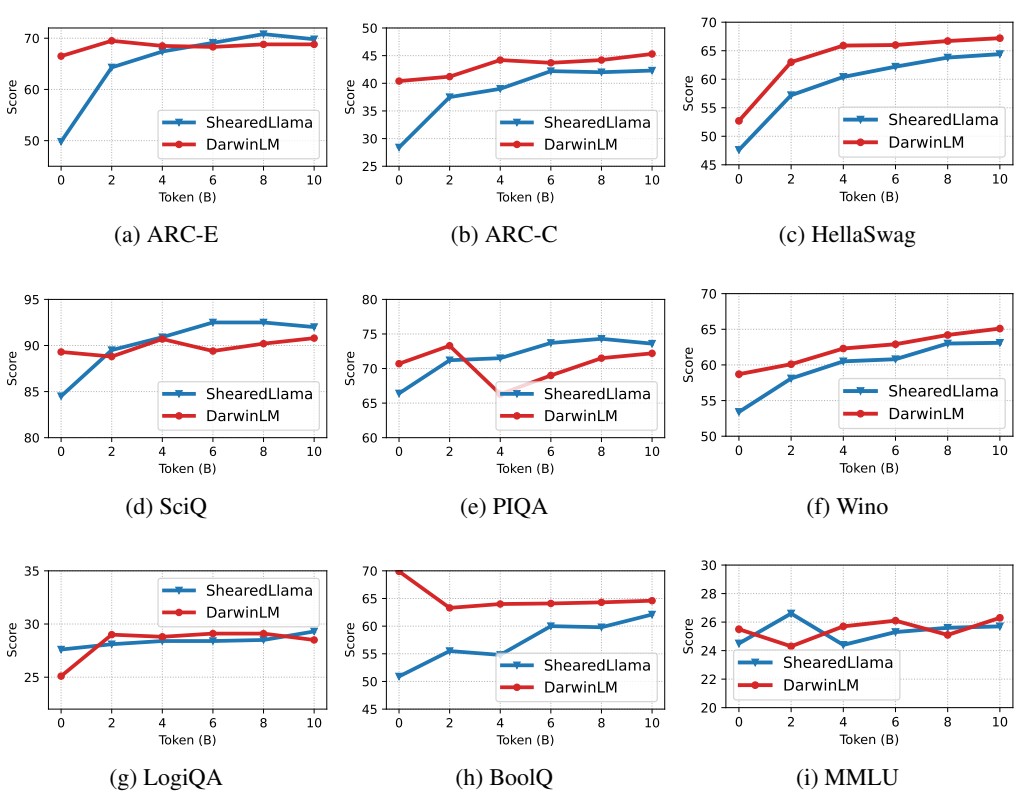

Figure 4: Post-training comparison of ShearedLlama and *DarwinLM* on each benchmark. Here, Llama-2-7B is pruned to 2.6B parameters via *DarwinLM*, and to 2.7B parameters with ShearedLlama. Both methods perform continued training on 10B tokens.

