# OpenReview forum: "DarwinLM: Evolutionary Structured Pruning of Large Language Model"
_ICLR.cc/2026/Conference — Submitted to ICLR 2026_

### Official Review · Reviewer_Va8F · 2025-10-31

**Soundness:** 4
**Presentation:** 4
**Contribution:** 4
**Rating:** 8
**Confidence:** 4

**Summary:**

This paper introduces DarwinLM, a training-aware structured pruning framework that treats model compression as an evolutionary process, where candidate submodels are iteratively generated, fine-tuned on small datasets, and selected based on fitness metrics. By integrating lightweight fine-tuning into the evolutionary search, DarwinLM effectively predicts long-term recovery potential and optimizes sparsity allocation across model layers and modules, including both dense and mixture-of-experts architectures. Experiments on Llama and Qwen models demonstrate that DarwinLM achieves state-of-the-art pruning efficiency, recovering over 90% of accuracy with only one-fifth of the training data required by prior methods while nearly doubling inference speed.

**Strengths:**

1. This paper is innovative in reducing the retraining cost of pruned models through evolutionary search, and it demonstrates strong practical value for real-world applications.

2. It is among the few works that conduct structured pruning on MoE models, which can inspire future research on lightweight optimization of MoE architectures.

3. The experiments are solid and comprehensive, demonstrating the effectiveness of the proposed method across several recent large language models.

4. The paper is clearly written and easy to follow.

**Weaknesses:**

The non-uniform sparsity allocation may affect the deployment efficiency of the pruned model. A more thorough discussion and analysis of this effect would strengthen the paper.

**Questions:**

Does the non-uniform sparsity allocation affect the deployment efficiency of the pruned model？

---

> ### Author Response · Authors · 2025-11-19
> **Response to Reviewer Va8F**
>
> Thank you for your detailed and constructive review. Here are our responses to your questions:
>
> W1 & Q1: The non-uniform sparsity allocation may affect the deployment efficiency of the pruned model. A more thorough discussion and analysis of this effect would strengthen the paper.
>
>
> Thank you for pointing this out. We provide the inference throughput and latency on vLLM inference framework and also add the comparison with Dense model and Shearedllama (with sequence length of 1024, request number of 128, single L40s GPUs):
>
> |     **Model**     | **Throughput (Tokens/s)** | **Latency (ms)** |
> |:-----------------:|:-------------------------:|:----------------:|
> |     llama2-7B     |          2469.57          |       51.83      |
> | Shearedllama-2.7B |          4482.95          |       28.55      |
> |   DarwinLM-2.7B   |          5675.29          |       22.55      |
>
> The results clearly show that the irregular shapes do not affect latency in a negative way. Moreover, we find that DarwinLM achieves higher throughput and lower latency compared with Shearedllama, at a similar parameter count. The reason DarwinLM is faster is that, with our approach, some Attention / MLP blocks are removed completely, which reduces both the computation, and the communication cost between SRAM and HBM inside the GPU. Furthermore, we believe that such a structure will bring extra efficiency benefits in the case of huge models, which require tensor-parallel or pipeline-parallel for inference, since removing a whole block significantly reduces block-wise communication cost.

---

### Official Review · Reviewer_iyCb · 2025-10-31

**Soundness:** 2
**Presentation:** 2
**Contribution:** 3
**Rating:** 4
**Confidence:** 3

**Summary:**

DarwinLM claims to improve structured pruning of LLMs using a hybrid of second-order saliency and evolutionary search over sparsity allocations. Each generation mutates layer-wise sparsity ratios, fine-tunes offspring, and keeps the fittest models. The authors report strong perplexity and downstream accuracy on LLaMA-2, LLaMA-3.1, and Qwen models, achieving up to 50–60% parameter reduction.

**Strengths:**

1. Targets structured pruning, which is the only kind of sparsity actually exploitable by modern inference frameworks.

2. Attempts to automate non-uniform sparsity allocation rather than relying on fixed heuristics.

3. The proposed approach could be extend to MoE-based model pruning which is a nice bonus.

**Weaknesses:**

1. The paper does not report latency, throughput, or kernel utilization results. I only found the authors roughly mentioned it in a small table (Table 4) Without demonstrating that the pruned models actually run faster on common inference engines such as TensorRT-LLM or vLLM, it remains unclear whether the proposed structured pruning translates into practical efficiency gains.

2. The evolutionary search involves training many fine-tuned offspring models, which likely requires substantial compute. However, the paper does not provide GPU-hour or data-usage accounting, making it difficult to assess whether the approach offers a favorable cost-benefit trade-off compared to simpler pruning baselines.

3. Combining second-order importance estimation with an evolutionary search procedure builds on several existing frameworks (e.g., AutoCompress, MetaPruning, ShearedLLaMA). The overall idea is coherent but does not introduce a clearly new optimization insight or theoretical advancement beyond prior art.

4. (minor) The Darwinian terminology somewhat overemphasizes the novelty of the approach. In essence, the method performs parameter mutation and selection within a standard hyper-parameter search loop rather than a biologically inspired or algorithmically distinct evolutionary process.

**Questions:**

1. Can you provide concrete latency numbers (e.g., ms/token) before and after pruning on e.g., A100/H100 GPUs under vLLM or TensorRT? Also, can you include more comparison methods instead of only the dense model as baselines in Table 4?

2. What is the full compute cost (GPU hours) of the evolutionary search versus the achieved inference savings?

3. Do your irregular sparsity patterns map cleanly to N:M kernels, or do they require custom CUDA kernels?

4. How does this differ, concretely, from FlexPrune or ShearedLLaMA beyond empirical tuning?

---

> ### Author Response · Authors · 2025-11-19
> **Response to Reviewer iyCb [1/2]**
>
> We would like to thank the reviewer for the detailed and constructive review. Here are our responses to your questions:
>
> > W1: The paper does not report latency, throughput, or kernel utilization results. I only found the authors roughly mentioned it in a small table (Table 4) Without demonstrating that the pruned models actually run faster on common inference engines such as TensorRT-LLM or vLLM, it remains unclear whether the proposed structured pruning translates into practical efficiency gains.
>
> Thank you for pointing this out. To address this question, we provide the inference throughput and latency on the state-of-the-art vLLM inference framework, and also add the detailed comparison with original Dense model and Shearedllama (with sequence length of 1024, request number of 128, single L40s GPUs):
>
> |     **Model**     | **Throughput (Tokens/s)** | **Latency (ms)** |
> |:-----------------:|:-------------------------:|:----------------:|
> |     llama2-7B     |          2469.57          |       51.83      |
> | Shearedllama-2.7B |          4482.95          |       28.55      |
> |   DarwinLM-2.7B   |          5675.29          |       22.55      |
>
> We find that DarwinLM achieves higher throughput and lower latency compared with Shearedllama, which has a similar number of parameters. The reason why DarwinLM is faster is that some Attention / MLP blocks are removed completely, which reduces both the computation, and the communication cost between SRAM and HBM inside the GPU. Furthermore, such a structure will bring extra efficiency benefits in huge models which require tensor-parallel or pipeline-parallel inference, since removing a whole block reduces communication as well.
>
> > W2: The evolutionary search involves training many fine-tuned offspring models, which likely requires substantial compute. However, the paper does not provide GPU-hour or data-usage accounting, making it difficult to assess whether the approach offers a favorable cost-benefit trade-off compared to simpler pruning baselines.
>
> This is a good point, but please note that we did provide a precise computational cost comparison relative to Shearedllama in Appendix, Table 15 (Table 19 in the revised version):
>
> | Model         | Hardware Requirement | Running Time |
> |---------------|----------------------|--------------|
> | ShearedLlama  | 8 × A100-80G         | 7.4h         |
> | DarwinLM      | 4 × L40S-48G         | 6.9h         |
>
> We find that DarwinLM costs about 2x less compared with ShearedLLama: this is because ShearedLLama requires extra training steps for the regularization process that is applied before pruning.
> Please also see the response to Reviewer 1oF1 Question 1 for an accuracy comparison relative to pure random search at the same computational budget.
>
> > W3: Combining second-order importance estimation with an evolutionary search procedure builds on several existing frameworks (e.g., AutoCompress, MetaPruning, ShearedLLaMA). The overall idea is coherent but does not introduce a clearly new optimization insight or theoretical advancement beyond prior art.
>
> While we clearly build upon prior work, our method leads to state-of-the-art results for structured pruning of LLMs. Our contribution lies in how these components are integrated for large-scale non-uniform LLM pruning. Specifically, DarwinLM introduces (1) a training-aware evolutionary search that incorporates lightweight fine-tuning during selection, improving correlation with final accuracy, and (2) our results show that this training-aware process is necessary for good results. We also extended our approach non-uniform pruning approach to Mixture-of-Experts models, which, to our knowledge, has not been previously explored.
>
> > W4: (minor) The Darwinian terminology somewhat overemphasizes the novelty of the approach. In essence, the method performs parameter mutation and selection within a standard hyper-parameter search loop rather than a biologically inspired or algorithmically distinct evolutionary process.
>
> We acknowledge this point: our intention was to use this terminology to describe the mutation and selection dynamics within our evolutionary search loop. Clearly, we do not claim to employ a biologically novel process.
>
> > Q1: Can you provide concrete latency numbers (e.g., ms/token) before and after pruning on e.g., A100/H100 GPUs under vLLM or TensorRT? Also, can you include more comparison methods instead of only the dense model as baselines in Table 4?
>
> We provided the throughput and latency in the response W1. We respectfully refer the reviewer to that answer.
>
> > Q2: What is the full compute cost (GPU hours) of the evolutionary search versus the achieved inference savings?
>
> We added the searching cost comparison in W2. Please have a look.

---

> > ### Author Response · Authors · 2025-11-19
> > **Response to Reviewer iyCb [2/2]**
> >
> > > Q3: Do your irregular sparsity patterns map cleanly to N:M kernels, or do they require custom CUDA kernels?
> >
> > Our method focuses on structured pruning, by which we mean removing rows/columns from the matrices. Therefore, no custom CUDA kernels are required to have the actual speedup, and our speedups would be realisable on any hardware. Our approach is compatible with N:M sparsity as well.
> >
> > > Q4: How does this differ, concretely, from FlexPrune or ShearedLLaMA beyond empirical tuning?
> >
> > Thank you for the question. First of all, we were unable to locate any published method or paper titled “FlexPrune”. We kindly ask the reviewer to provide the reference of FlexPrune so that we can review it and ensure an accurate comparison.  Apart from this, our work differs from Sheared‑LLaMA in three main ways. First, we incorporate evolutionary search over non-uniform layer/width allocations, rather than selecting a fixed target shape a priori. Second, we introduce training-aware selection, where each candidate model receives a lightweight fine-tuning pass during the search process before final selection.  Finally, we demonstrate the method on larger LLMs (e.g., 8B- to 70B-parameter models) and show improved data efficiency (e.g., using 5 \times less post-pruning tokens) compared to Sheared-LLaMA.

---

### Official Review · Reviewer_6cfz · 2025-11-01

**Soundness:** 3
**Presentation:** 2
**Contribution:** 2
**Rating:** 4
**Confidence:** 4

**Summary:**

DarwinLM proposes training-aware, non-uniform structured pruning for LLMs. It first builds a per-layer “sparsity level” database via second-order one-shot pruning, then uses an evolutionary search with “level-switch” mutations under a target size/speed constraint. Fitness is measured via KL divergence to the dense model. Selection is multi-step and training-aware, followed by a final 10B-token post-compression finetune. The method reports one-shot and post-train results on Llama-2-7B, Llama-3.1-8B, Qwen-2.5-14B, and an MoE case (Qwen-3-30B-A3B).

**Strengths:**

* Clear pipeline: second-order one-shot pruning, sparsity-level database, evolutionary search with speed/size constraints, short finetune.

* Training-aware selection nicely predicts which offspring recover best after longer finetunes; ablation supports the idea.

**Weaknesses:**

* In the main table (Table 1), the performance gaps shrink notably after finetuning. It’s unclear whether DarwinLM’s one-shot advantage persists under longer training or on performance after training converges. The pruned+finetuned models may still underperform public small dense models (e.g., pruned-llama 3.1 8B vs llama 3.2 3B model). While this is reasonable as only 10B tokens are used for recovering performance, this still raise the need for further fine-tuning the models to be actually used. It would be helpful to include results under longer training or distillation to see if the initialization gain is still there.

* The search can yield highly non-uniform (unbalanced) per-layer sparsity. It would be beneficial to show how the final architecture looks and whether unbalanced shapes can harm optimization or stability during continued training. Comparing the final shape with Shearedllama can provide more insights.

* The paper states that "this is the first work to explore structured pruning in MoE architectures" in the abstract, which is not accurate and overstated. Some works have worked on structured pruning of MoE [1,2].

* It's not very accurate to claim "Orthogonal to Minitron/Flextron”. Those pipelines couple structured pruning with KD and depth/width choices; DarwinLM’s search over non-uniform width allocations overlaps in spirit.

* The non-uniform architecture can also affect the inference efficiency. Irregular shapes can reduce kernel efficiency and complicate inference optimization. Some discussion or experiments comparing uniform and non-uniform will be helpful.


[1] SlimMoE: Structured Compression of Large MoE Models via Expert Slimming and Distillation

[2] Demystifying the Compression of Mixture-of-Experts Through a Unified Framework

**Questions:**

Included above.

---

> ### Author Response · Authors · 2025-11-19
> **Response to Reviewer 6cfz [1/2]**
>
> Thank you for your concrete and constructive review. Here are our responses to your questions:
>
> > W1: In the main table (Table 1), the performance gaps shrink notably after finetuning. It’s unclear whether DarwinLM’s one-shot advantage persists under longer training or on performance after training converges. The pruned+finetuned models may still underperform public small dense models (e.g., pruned-llama 3.1 8B vs llama 3.2 3B model). While this is reasonable as only 10B tokens are used for recovering performance, this still raise the need for further fine-tuning the models to be actually used. It would be helpful to include results under longer training or distillation to see if the initialization gain is still there.
>
> Thank you for pointing this out. Please note that the Llama 3.2 3B model you mentioned is extremely over-trained via distillation, over more than 9T tokens. So it is unfair to compare it directly to our compressed model, which uses _two orders of magnitude_ less data for tuning. For a fairer comparison, please examine our OLMO accuracy comparison, which is a model trained with open data (similar to our fine-tuning setup) in the appendix, in Table 10 (Table 14 in the revised version). We also repeat the detailed results here:
>
> | Model (Training token)      | SciQ | PIQA | WG  | ArcE | ArcC | HS  | LogiQA | BoolQ | MMLU | Avg  |
> |-----------------------------|------|------|-----|------|------|-----|--------|-------|------|------|
> | OLMO 7B (2.5T)              | 92.8 | 79.4 | 70.4| 73.3 | 44.9 | 77.1| 27.9   | 72.5  | 28.3 | 62.9 |
> | *DarwinLM* 4.6B (10B)       | 93.2 | 74.8 | 67.4| 73.2 | 51.6 | 71.3| 30.7   | 71.1  | 40.6 | 63.7 |
> | Baichuan2 7B (2.6T)         | 94.8 | 77.1 | 72.2| 75.0 | 49.5 | 73.0| 28.7   | 73.9  | 54.0 | 66.4 |
> | OLMO 0424 7B (2T)           | 96.1 | 80.1 | 72.1| 73.8 | 49.2 | 78.0| 29.3   | 80.8  | 52.1 | 67.9 |
> | *DarwinLM* 8.4B (10B)       | 89.5 | 78.1 | 70.7| 79.6 | 57.6 | 74.9| 33.5   | 73.9  | 57.9 | 68.4 |
>
> With less training data and model parameters, DarwinLM outperforms OLMO 7B, which is trained with 2.5T tokens.
>
> To further address your question, we provide results with more training tokens (50B) on our MoE models, and compare accuracies with similar-sized models:
>
> |              Methods              | SciQ | PIQA |  WG  | ARC-E | ARC-C |  HS  | LogiQA | BoolQ | MMLU |  Avg |
> |:---------------------------------:|:----:|:----:|:----:|:-----:|:-----:|:----:|:------:|:-----:|:----:|:----:|
> |               Dense               |  97  | 79.7 | 71.5 |  79.7 |  68.6 | 77.8 |  34.7  |  88.8 | 79.6 | 75.2 |
> |  DeepSeek MoE base 16A2B 2T token | 92.9 | 80.5 | 72.7 |  75.9 |  53.2 | 79.9 |  29.1  |  72.9 |  45  | 66.9 |
> | DeepSeek-V2-Lite 16A2B 5.7T token | 93.5 |  79  | 69.2 |  75.5 |  51.9 | 74.6 |  29.1  |  74.3 | 48.4 | 66.1 |
> |    DarwinLM 16A2B MoE 10B token   | 95.9 | 76.2 | 69.4 |  80.4 |   59  | 69.9 |  32.5  |   77  | 66.9 | 69.7 |
> |    DarwinLM 16A2B MoE 50B token   |  96  | 77.1 | 70.1 |  81.9 |  60.5 | 72.5 |  32.7  |   78  | 69.1 | 70.8 |
>
> Please observe that, DarwinLM achieves better performance than DeepSeek MoE base and DeepSeek-V2-Lite, which are trained with 2T and 5.7T tokens, respectively.

---

> > ### Author Response · Authors · 2025-11-19
> > **Response to Reviewer 6cfz [2/2]**
> >
> > >W2: The search can yield highly non-uniform (unbalanced) per-layer sparsity. It would be beneficial to show how the final architecture looks and whether unbalanced shapes can harm optimization or stability during continued training. Comparing the final shape with Shearedllama can provide more insights.
> >
> > |            Type            |                                                                                                                                                                                             |   |
> > |:--------------------------:|:-------------------------------------------------------------------------------------------------------------------------------------------------------------------------------------------:|---|
> > |   DarwinLM Attn Head Num   |                                    25, 21, 18, 18, 14, 10, 14, 10, 18, 14, 18, 0, 0, 28, 21, 10, 18, 14, 10, 18, 10, 10, 10, 14, 0, 0, 14, 1, 4, 0, 6, 0                                    |   |
> > | Shearedllama Attn Head Num |                                                                                      20 for all layers                                                                                      |   |
> > |      DarwinLM MLP Size     | 3104, 8032, 6496, 4256, 5280, 5280, 4256, 3104, 5280, 4256, 1824, 0, 3104, 5280, 5280, 4256, 4256, 6496, 6496, 5280, 3104, 4256, 4256, 3104, 3104, 3104, 4256, 3104, 3104, 3104, 6496, 6496 |   |
> > |    Shearedllama MLP Size   |                                                                                     6912 for all layers                                                                                     |   |
> >
> > We provide the detailed structure information comparison with ShearedLlama. Shearedllama is an uniformly pruned model while DarwinLM owns different sparsities for different blocks. We did not detect the instability that is suggested in W1: specifically, the non-uniformly pruned model consistently improves with more training tokens.
> >
> > > W3: The paper states that "this is the first work to explore structured pruning in MoE architectures" in the abstract, which is not accurate and overstated. Some works have worked on structured pruning of MoE [1,2].
> >
> > Thank you for bringing this up. We will amend our claim regarding the applications to MoEs to clarify that we are “the first work to explore non-uniform structured pruning in MoE architectures”. We will cite the papers you mentioned as concurrent work and integrate them into our related work discussion.
> >
> > > W4: It's not very accurate to claim "Orthogonal to Minitron/Flextron”. Those pipelines couple structured pruning with KD and depth/width choices; DarwinLM’s search over non-uniform width allocations overlaps in spirit.
> >
> > We acknowledge this point: what we wanted to say is that we focus primarily on _improved pruning techniques_: from this perspective, our method is orthogonal to the training strategies introduced in works such as Minitron/Flextron. We will revise our presentation on this point to make it clearer.
> >
> >
> > > W5: The non-uniform architecture can also affect the inference efficiency. Irregular shapes can reduce kernel efficiency and complicate inference optimization. Some discussion or experiments comparing uniform and non-uniform will be helpful.
> >
> > This is a great point, but this does not affect our claims. Specifically, we provide the inference throughput and latency on the leading vLLM open-source inference engine, and also add comparisons with the baseline model and with Shearedllama (with sequence length of 1024, request number of 128, on single L40s GPUs):
> >
> > |     **Model**     | **Throughput (Tokens/s)** | **Latency (ms)** |
> > |:-----------------:|:-------------------------:|:----------------:|
> > |     llama2-7B     |          2469.57          |       51.83      |
> > | Shearedllama-2.7B |          4482.95          |       28.55      |
> > |   DarwinLM-2.7B   |          5675.29          |       22.55      |
> >
> > The results clearly show that the irregular shapes do not affect latency in a negative way. Moreover, we find that DarwinLM achieves higher throughput and lower latency compared with Shearedllama, at a similar parameter count. The reason DarwinLM is faster is that, with our approach, some Attention / MLP blocks are removed completely, which reduces both the computation, and the communication cost between SRAM and HBM inside the GPU. Furthermore, we believe that such a structure will bring extra efficiency benefits in the case of huge models, which require tensor-parallel or pipeline-parallel for inference, since removing a whole block significantly reduces block-wise communication cost.

---

### Official Review · Reviewer_1oF1 · 2025-11-03

**Soundness:** 3
**Presentation:** 3
**Contribution:** 3
**Rating:** 6
**Confidence:** 2

**Summary:**

This paper introduces DarwinLM, a method for structured pruning of large language models that uses evolutionary search to find optimal non-uniform compression patterns. The core idea is to generate multiple "offspring" models through mutations that shift sparsity between layers, then select the best candidates using a multi-step training-aware process. The method builds a database of pre-pruned layers at different sparsity levels using second-order information, then searches over combinations of these levels while incorporating lightweight fine-tuning to predict which models will perform best after full training. The authors test on several models (Llama-2-7B, Llama-3.1-8B, Qwen-2.5-14B) and show improvements over uniform pruning and competing methods like ShearedLlama and ZipLM. They claim to achieve comparable or better accuracy while using 5x less training data than ShearedLlama. The method is also extended to MoE architectures, which is claimed as a first for structured pruning.

**Strengths:**

1. The multi-step selection strategy that progressively increases fine-tuning data (10K→50K→100K→200K tokens) is both intuitive and empirically validated. By showing in Figure 2 that small-scale training predicts larger-scale performance, the paper introduces a practical and well-motivated solution to a long-standing inefficiency in pruning and neural architecture search.

2. The method is evaluated across diverse model families and sizes—up to 70B parameters—with generally fair baselines and detailed ablations (offspring count, sparsity levels, fitness metrics). The results consistently demonstrate superiority over baselines like ShearedLlama and ZipLM.

**Weaknesses:**

See Questions.

**Questions:**

1. Evolutionary search feels overcomplicated. The core is just trying different per-layer sparsity combinations. The mutation operator is trivial (swap sparsity between two layers).  Do simpler search methods like beam search also work? It would be better to have comparison with random search given the same budget.

2. Search cost are unclear. 200 generations × 16 offspring = 3,200 evaluations, each requiring model stitching and training. How does total cost compare to ZipLM's dynamic programming or ShearedLlama's approach?

3. How sensitive is DarwinLM to the choice of training-aware selection token budgets (10K–200K)? Could different scaling change the final selection outcome?

---

> ### Author Response · Authors · 2025-11-19
> **Response to Reviewer 1oF1**
>
> Thank you for your detailed and constructive review. Here are our responses to your questions:
>
> > Q1: Evolutionary search feels overcomplicated. The core is just trying different per-layer sparsity combinations. The mutation operator is trivial (swap sparsity between two layers). Do simpler search methods like beam search also work? It would be better to have comparison with random search given the same budget.
>
> First, there is evidence that complex search procedures may be _necessary_ for good results in the case of structured compression, as the corresponding loss landscapes tend to be very “spiky”. For instance, the EvoPress [1] paper gives examples of block dropping where accuracy is not even monontonic w.r.t. compression: a _more compressed_ one-shot configuration can be more accurate than a _less compressed_ one.
> ZipLM is also a search technique, which works by dynamic programming with a sparsity constraint, and assumes loss monotonicity (i.e. less compression is always more accurate than more compression).
>
> To fully address your concern, we present experimental results for 1) Random Search, 2) ZipLM, and 3) DarwinLM for the Llama-3.1-8B model, below. (The experiments follow our standard one-shot setup from the paper.)
>
> | Model        | Method        | Param. | SciQ | PIQA | WG  | ARC-E | ARC-C | HS  | LogiQA | BoolQ | MMLU | Avg  |
> |--------------|---------------|--------|------|------|-----|-------|-------|-----|--------|-------|------|------|
> | Llama-3.1-8B | Random Search | 4.6B   | 78.1 | 65.5 | 52.3| 54.5  | 26.2  | 31.6| 24.1   | 62.1  | 26.5 | 46.7 |
> |              | ZipLM         | 6B     | 65.5 | 60.6 | 56.0| 40.2  | 36.2  | 34.4| 28.1   | 63.0  | 27.9 | 45.7 |
> |              | *DarwinLM (one-shot)* | 4.6B   | 84.9 | 69.4 | 57.3| 59.6  | 34.2  | 44.6| 24.1   | 62.2  | 28.5 | 51.6 |
>
> The results show a major accuracy advantage in favor of DarwinLM, with an improvement of almost 4% on average, across tasks, relative to Random search, and even higher relative to ZipLM. This highlights the advantage of our search strategy, and also validates our earlier point that random search is _necessary_ for good results.
>
> > Q2: Search cost are unclear. 200 generations × 16 offspring = 3,200 evaluations, each requiring model stitching and training. How does total cost compare to ZipLM's dynamic programming or ShearedLlama's approach?
>
> Please note that we had a detailed cost comparison relative to Shearedllama in the Appendix, specifically Table 15 (Table 19 in the revised version). We repeat it here:
>
> | Model         | Hardware Requirement | Running Time |
> |---------------|----------------------|--------------|
> | ShearedLlama  | 8 × A100-80G         | 7.4h         |
> | DarwinLM      | 4 × L40S-48G         | 6.9h         |
>
> In summary, DarwinLM costs about 2x less for searching compared with ShearedLLama, since ShearedLLama requires extra fine-tuning for the pruning/regularization process itself.
>
> > Q3:  How sensitive is DarwinLM to the choice of training-aware selection token budgets (10K–200K)? Could different scaling change the final selection outcome?
>
> Please note that we have already included the detailed ablation and discussion regarding to the selection tokens in Appendix Table 19 (Table 23 in the revised version). We repeat the results here:
>
> | Finetuning Tokens | SciQ | PIQA | WG  | ArcE | ArcC | HS  | LogiQA | BoolQ | Avg  |
> |-------------------|------|------|-----|------|------|-----|--------|-------|------|
> | [10K, 50K, 100K]  | 84.9 | 69.4 | 57.3| 59.6 | 34.2 | 44.6| 24.1   | 62.2  | 51.6 |
> | [5K, 10K, 20K]    | 85.8 | 69.8 | 56.1| 60.9 | 33.6 | 43.8| 25.3   | 61.1  | 51.6 |
>
> Please observe that, with different numbers of finetuning tokens, the search results are generally similar.
>
> [1] Sieberling O, Kuznedelev D, Kurtic E, et al. Evopress: Towards optimal dynamic model compression via evolutionary search[J]. arXiv preprint arXiv:2410.14649, 2024.

---

### Author Response · Authors · 2025-11-27
**General Response**

We sincerely thank the AC and all reviewers for the constructive feedback and thoughtful suggestions. We are encouraged by the positive comments from reviewers, particularly for its innovative training-aware evolutionary search **[Reviewer iyCb, Va8F]**, effective non-uniform structured pruning strategy **[Reviewer 1oF1, 6cfz]**, strong empirical results across diverse models **[Reviewer 1oF1, 6cfz, iyCb, Va8F]**, and clear practical value in reducing retraining cost while maintaining compatibility with real-world inference frameworks **[Reviewer iyCb]**.

We have provided the response to your concerns. We kindly ask you to review our rebuttal and if possible, share any further thoughts or questions you may have. Your insights are greatly appreciated and will help us improve the quality of our work.

We have carefully revised our final uploaded manuscript in accordance with the comments provided, marked in ***blue***. Specifically, we have incorporated the following updates:

- Added comparisons against Random Search and ZipLM to demonstrate the necessity and advantage of our evolutionary search strategy, in the Appendix, Table 8. **[Reviewer 1oF1 Q1]**
- Added extensive results demonstrating that DarwinLM's performance advantage persists and even grows with longer fine-tuning (up to 50B tokens), outperforming models trained with orders of magnitude more data, in the Appendix, Table 9. **[Reviewer 6cfz W1]**
- Provided a detailed structural comparison with ShearedLLaMA, showcasing the non-uniform sparsity patterns of DarwinLM and confirming their stability during training, in the Appendix, Table 10. **[Reviewer 6cfz W2]**
- Revised the abstract and related work section to accurately reflect the state of the art regarding MoE pruning, acknowledging concurrent works. **[Reviewer 6cfz W3]**
- Clarified the relationship between DarwinLM and methods like Minitron/Flextron, emphasizing our focus on improved pruning techniques in related work. **[Reviewer 6cfz W4]**
- Added thorough inference efficiency analysis using the vLLM framework, reporting concrete throughput and latency numbers. These results show that DarwinLM's non-uniform structure does not harm, but actually improves, inference speed compared to uniform baselines like ShearedLLaMA, in the Appendix, Table 7. **[Reviewer 6cfz W5, Reviewer iyCb W1, Reviewer Va8F W1 & Q1]**

We deeply appreciate the reviewers' invaluable feedback, which has greatly helped enhance the quality, clarity, and rigor of our work. Thank you for your time and effort throughout the review process.

---

### Author Response · Authors · 2025-12-03
**Message to AC**

We thank AC and all reviewers for your time on our work. Due to unexpected issues in the review process, we were unable to receive final responses to our rebuttal from all reviewers. Here, we briefly summarize our reviews and rebuttal.  Overall, the reviewers recognize the strengths, novelty, and practical impact of DarwinLM. Three reviewers (**1oF1, 6cfz, Va8F**) explicitly praise the method’s clear pipeline, strong empirical results, and practical value, while **Reviewer Va8F** rates the paper as **“accept, good paper”** with excellent scores across all criteria.

***Key Contributions Highlighted by Reviewers***

- Innovative training-aware structured pruning framework integrating second-order one-shot pruning with evolutionary search (**1oF1, 6cfz, iyCb, Va8F**).
- Strong empirical performance with consistent improvements across LLaMA and Qwen models (**1oF1, Va8F**).
- Reduced retraining cost and effective prediction of long-term recovery through lightweight fine-tuning within the search loop (**1oF1, Va8F**).
- Extension to MoE pruning, one of the first works to do so systematically (**1oF1, Va8F**).
- Clear writing and comprehensive experiments (**Va8F**).

Across all concerns, we provided direct empirical evidence, clarified methodology, and added comparisons, addressing reviewers’ questions thoroughly:

1. Search complexity and necessity of evolutionary search (**Reviewer 1oF1, iyCb**)

We conducted new experiments comparing Random Search, ZipLM, and DarwinLM, showing that DarwinLM outperforms random search by ~4% and ZipLM by ~6% on average, demonstrating that evolutionary search is not only non-trivial but necessary for good performance, as shown:

| Model        | Method        | Param. | SciQ | PIQA | WG  | ARC-E | ARC-C | HS  | LogiQA | BoolQ | MMLU | Avg  |
|--------------|---------------|--------|------|------|-----|-------|-------|-----|--------|-------|------|------|
| Llama-3.1-8B | Random Search | 4.6B   | 78.1 | 65.5 | 52.3| 54.5  | 26.2  | 31.6| 24.1   | 62.1  | 26.5 | 46.7 |
|              | ZipLM         | 6B     | 65.5 | 60.6 | 56.0| 40.2  | 36.2  | 34.4| 28.1   | 63.0  | 27.9 | 45.7 |
|              | *DarwinLM (one-shot)* | 4.6B   | 84.9 | 69.4 | 57.3| 59.6  | 34.2  | 44.6| 24.1   | 62.2  | 28.5 | 51.6 |


2. Computational cost of evolutionary search (**Reviewer 1oF1, iyCb**)

We originally provided detailed GPU-hour comparisons with ShearedLLaMA in the Appendix. The search cost of DarwinLM is ~2$\times$ lower than ShearedLLaMA’s due to eliminating additional fine-tuning stages required by their method.

3. Sensitivity to fine-tuning token budgets (**Reviewer 1oF1**)

We originally provided experiments showing that final performance is stable across token budgets (5K–200K) in the Appendix, demonstrating the robustness of our training-aware selection process.

4. Non-uniform sparsity and potential instability (**Reviewer 6cfz**)

We added detailed architecture comparisons and long-training experiments. We don’t observe any training instability and performance improves with more training tokens, as shown:

|              Methods              | SciQ | PIQA |  WG  | ARC-E | ARC-C |  HS  | LogiQA | BoolQ | MMLU |  Avg |
|:---------------------------------:|:----:|:----:|:----:|:-----:|:-----:|:----:|:------:|:-----:|:----:|:----:|
|               Dense               |  97  | 79.7 | 71.5 |  79.7 |  68.6 | 77.8 |  34.7  |  88.8 | 79.6 | 75.2 |
|  DeepSeek MoE base 16A2B 2T token | 92.9 | 80.5 | 72.7 |  75.9 |  53.2 | 79.9 |  29.1  |  72.9 |  45  | 66.9 |
| DeepSeek-V2-Lite 16A2B 5.7T token | 93.5 |  79  | 69.2 |  75.5 |  51.9 | 74.6 |  29.1  |  74.3 | 48.4 | 66.1 |
|    DarwinLM 16A2B MoE 10B token   | 95.9 | 76.2 | 69.4 |  80.4 |   59  | 69.9 |  32.5  |   77  | 66.9 | 69.7 |
|    DarwinLM 16A2B MoE 50B token   |  96  | 77.1 | 70.1 |  81.9 |  60.5 | 72.5 |  32.7  |   78  | 69.1 | 70.8 |


5. Latency and throughput (**Reviewer iyCb, Va8F**)

We added comprehensive inference benchmarks. Non-uniformity does not harm inference efficiency; in fact, DarwinLM demonstrates higher throughput and lower latency than ShearedLLaMA on vLLM, as shown:

|     **Model**     | **Throughput (Tokens/s)** | **Latency (ms)** |
|:-----------------:|:-------------------------:|:----------------:|
|     llama2-7B     |          2469.57          |       51.83      |
| Shearedllama-2.7B |          4482.95          |       28.55      |
|   DarwinLM-2.7B   |          5675.29          |       22.55      |

***Summary***

The reviewers acknowledge the strength, novelty, and comprehensiveness of our work. All major concerns, such as computational cost, necessity of evolutionary search, inference efficiency and sensitivity analyses, have been fully addressed through additional experiments, detailed comparisons, and clarifications. Thank AC again for your reviewing efforts on our work.

Best,

The Authors

---

### Meta-Review · Program_Chairs · 2026-01-09

**Summary:**

The paper proposes a training-aware structured pruning framework using evolutionary search and presents extensive empirical results across several dense and MoE models. Reviewers generally acknowledge the clarity of the pipeline and the practical motivation of reducing retraining cost, and one reviewer highlights strong empirical performance.

However, several core concerns remain unresolved under a conservative reading. Key questions about isolating the architectural contribution from training and data effects persist, particularly regarding whether reported gains would remain after longer training or at convergence for dense models. Requests for simpler or more direct baselines (e.g., beam search) and clearer, normalized accounting of search cost relative to comparable methods remain unaddressed. Some claims rely heavily on added experiments without explicit reviewer confirmation, and aspects of reproducibility and deployment benchmarking (e.g., broader hardware settings) are still incomplete.

Overall, the initial reviews reflect mixed opinions and marginal scores, including explicit statements that the paper could reasonably go either way, and no reviewer provided a clear post-rebuttal signal indicating a score increase. Given the interrupted review process, this assessment represents a best-effort, conservative judgment based solely on the available materials and discussion, without assuming any score changes.

**Reviewer Concerns:**

Reviewer Concerns*

Reviewer 1oF1:

* Addressed:
  • Random search and ZipLM comparison added; shows clear accuracy gains.
  • Token-budget sensitivity analyzed; results appear robust.
* Outstanding:
  • No beam-search comparison as explicitly requested.
  • No direct search-cost comparison against ZipLM.

Reviewer 6cfz:

* Addressed:
  • Overstated MoE novelty claim corrected.
  • Inference efficiency of non-uniform sparsity supported by benchmarks.
* Outstanding:
  • Lack of clear dense-model evidence that gains persist after full convergence or distillation.

Reviewer iyCb:

* Addressed:
  • Clarified no custom CUDA kernels are required.
* Outstanding:
  • Incomplete deployment benchmarks (e.g., A100/H100, TensorRT-LLM).
  • Incomplete end-to-end compute cost–benefit analysis.

Reviewer Va8F:

* Addressed:
  • Deployment efficiency concern addressed with throughput/latency results.
* Outstanding:
  • None.

**Reviewer Scores:**

Reviewer 1oF1:

Original score: 6

Likely post-rebuttal score: 6

Justification:
• No explicit reviewer signal indicating a score change.
• Outstanding major concerns remain (beam-search comparison, ZipLM cost).

Reviewer 6cfz:

Original score: 4

Likely post-rebuttal score: 4

Justification:
• No explicit post-rebuttal signal from the reviewer.
• Major concerns on long-training convergence for dense models remain only partially addressed.

Reviewer iyCb:

Original score: 4

Likely post-rebuttal score: 4

Justification:
• No explicit reviewer indication of improved assessment.
• Outstanding concerns on deployment benchmarks and cost–benefit analysis remain.

Reviewer Va8F:

Original score: 8

Likely post-rebuttal score: 8

Justification:
• No explicit reviewer signal for change.
• Main concern on deployment efficiency was addressed, but conservative default is no change.

---

### Decision · Program_Chairs · 2026-01-26

Reject